# Histopathology images predict multi-omics aberrations and prognoses in colorectal cancer patients

Pei-Chen Tsai [1,2], Tsung-Hua Lee[2], Kun-Chi Kuo[2], Fang-Yi Su[2], Tsung-Lu Michael Lee [3], Eliana Marostica [1,4], Tomotaka Ugai[5,6], Melissa Zhao[6], Mai Chan Lau[6], Juha P. Väyrynen [7], Marios Giannakis[8], Yasutoshi Takashima[6], Seyed Mousavi Kahaki[6], Kana Wu[9], Mingyang Song[5], Jeffrey A. Meyerhardt[8], Andrew T. Chan [10,11], Jung-Hsien Chiang [2] ✉, Jonathan Nowak[6,13], Shuji Ogino [5,6,12,13] & Kun-Hsing Yu [1,6,13] ✉

Histopathologic assessment is indispensable for diagnosing colorectal cancer (CRC). However, manual evaluation of the diseased tissues under the microscope cannot reliably inform patient prognosis or genomic variations crucial for treatment selections. To address these challenges, we develop the Multi-omics Multi-cohort Assessment (MOMA) platform, an explainable machine learning approach, to systematically identify and interpret the relationship between patients' histologic patterns, multi-omics, and clinical profiles in three large patient cohorts ($n = 1888$). MOMA successfully predicts the overall survival, disease-free survival (log-rank test $P$-value<0.05), and copy number alterations of CRC patients. In addition, our approaches identify interpretable pathology patterns predictive of gene expression profiles, microsatellite instability status, and clinically actionable genetic alterations. We show that MOMA models are generalizable to multiple patient populations with different demographic compositions and pathology images collected from distinctive digitization methods. Our machine learning approaches provide clinically actionable predictions that could inform treatments for colorectal cancer patients.

Colorectal cancer (CRC) is the second most common cause of cancer death in the United States, accounting for nearly 53,000 deaths annually[1]. Histopathologic evaluation remains a cornerstone for diagnosing and staging CRC, and the histology subtypes and genetic variations are the keys to treatment selection[2]. However, inter-rater variability in histopathology diagnoses has been reported[2, 3], and the genomic profiling process requires days to weeks to complete and is not available to every hospital in the developing world. These

[1]Department of Biomedical Informatics, Harvard Medical School, Boston, MA, USA. [2]Department of Computer Science and Information Engineering, National Cheng Kung University, Tainan, Taiwan ROC. [3]Department of Computer Science and Information Engineering, Southern Taiwan University of Science and Technology, Tainan, Taiwan ROC. [4]Division of Health Sciences and Technology, Harvard-Massachusetts Institute of Technology, Boston, MA, USA. [5]Department of Epidemiology, Harvard T.H. Chan School of Public Health, Boston, MA, USA. [6]Department of Pathology, Brigham and Women's Hospital, Boston, MA, USA. [7]Cancer and Translational Medicine Research Unit, Medical Research Center Oulu, Oulu University Hospital and University of Oulu, Oulu, Finland. [8]Department of Medicine, Dana Farber Cancer Institute, Boston, MA, USA. [9]Department of Nutrition, Harvard T.H. Chan School of Public Health, Boston, MA, USA. [10]Department of Medicine, Massachusetts General Hospital, Boston, MA, USA. [11]Channing Division of Network Medicine, Department of Medicine, Brigham and Women's Hospital, Boston, MA, USA. [12]Broad Institute of MIT and Harvard, Cambridge, MA, USA. [13]These authors jointly supervised this work: Jonathan Nowak, Shuji Ogino, Kun-Hsing Yu. ✉e-mail: jchiang@mail.ncku.edu.tw; Kun-Hsing_Yu@hms.harvard.edu

**Table 1 | Patient characteristics of our study cohorts**

| Patient characteristics | | TCGA | NHS-HPFS | PLCO |
|---|---|---|---|---|
| **Number of patients** | | *N* = 628 | *N* = 927 | *N* = 333 |
| Age (Standard Deviation) | | 66.3 ± 12.8 | 62.4 ± 9.6 | 65.0 ± 4.7 |
| Sex | Male | 334 (53.2%) | 413 (44.6%) | 213 (64.0%) |
| | Female | 294 (46.8%) | 512 (55.3%) | 120 (36.0%) |
| Race | Not Available | 255 (40.61%) | 390 (42.07%) | 175 (52.6%) |
| | Black or African American | 65 (10.35%) | 8 (0.86%) | 8 (2.4%) |
| | White | 295 (46.97%) | 526 (56.74%) | 120 (36.0%) |
| | Asian | 12 (1.91%) | 3 (0.32%) | 26 (7.8%) |
| | Native American or Alaska Native | 1 (0.16%) | 0 (0%) | 0 (0.0%) |
| | Pacific Islander | 0 (0.0%) | 0 (0%) | 4 (1.2%) |
| Tumor Location | Proximal Colon | 258 (42.5%) | 469 (50.4%) | 127 (38.1%) |
| | Distal Colon | 185 (30.5%) | 280 (30.1%) | 88 (26.4%) |
| | Rectum | 164 (27.0%) | 181 (19.5%) | 118 (35.4%) |
| Disease Stage | Stage I | 108 (17.2%) | 198 (21.4%) | 49 (14.7%) |
| | Stage II | 229 (36.5%) | 281 (30.3%) | 64 (19.2%) |
| | Stage III | 181 (28.8%) | 248 (26.8%) | 50 (15.0%) |
| | Stage IV | 90 (14.3%) | 134 (14.5%) | 16 (4.8%) |
| | Unknown | 20 (3.2%) | 66 (7.1%) | 154 (46.2%) |
| MSI | High | 65 (14.3%) | 150 (16.7%) | - |
| | Low/negative | 389 (85.7%) | 750 (83.3%) | - |
| *BRAF* mutation* | *BRAF* mutation in any loci | 62 (10.4%) | 136 (15.0%) | - |
| | *BRAF* c.1799T > A (p.V600E) mutation | 48 (8.32%) | - | - |
| | Wild-Type | 529 (89.5%) | 770 (85.0%) | - |
| CIMP | High | 58 (12.8%) | 155 (18.1%) | - |
| | Low/negative | 396 (87.2%) | 703 (81.9%) | - |

*Gene names are italicized.

limitations have hindered CRC patients from receiving timely and appropriate treatments.

With the recent development of reliable whole-slide pathology scanners and high-performing computer vision techniques, quantitative pathology evaluation has become increasingly feasible[4]. Several studies using machine learning techniques reported remarkable diagnostic accuracy for various cancer types, such as lung, breast, ovarian, renal cell, and colorectal carcinomas[5–11]. Previous works also demonstrated unexpected correlations between histopathology image features and clinically actionable molecular variations, such as microsatellite instability and *PTEN* gene deletion, in colorectal carcinoma samples[12, 13]. These studies indicate that high-resolution pathology images contain underutilized biomedical signals useful for personalizing cancer care[14–19].

Nonetheless, many computational challenges hinder the extraction of useful histopathology signals, and several reports expressed concerns about the generalizability of deep learning models[20]. Typical high-resolution digital pathology whole-slide images of colorectal cancer tissues contain up to billions of pixels, making it infeasible for standard convolutional neural networks to process the whole image at once. In addition, deep learning models are highly complex, and it is difficult to connect the image patterns discovered by these data-driven models with biological knowledge[21]. Furthermore, since there are a large number of parameters that researchers need to optimize in data-driven machine learning models, generalizability to other image acquisition methods remains a substantial challenge to many digital

pathology models[22]. The lack of extensive validation in different patient cohorts can diminish the applicability of machine learning models in clinical settings.

In this study, we propose the Multi-omics Multi-cohort Assessment (MOMA) system, an explainable machine learning framework for analyzing digital pathology images at scale. Our informatics methods successfully predict the prognoses of early-stage colorectal cancer patients and achieve state-of-the-art performance in identifying the genomics and proteomics status of cancer samples using a weakly supervised prediction framework. We connect high-resolution digital pathology images with clinically actionable multi-omics aberrations, and we identify interpretable pathology predictors of patients' survival outcomes. We further validate our framework in multiple large patient cohorts and demonstrate its generalizability in different populations and using different image acquisition methods. Our study provides a robust and flexible machine learning framework for scalable histopathology image analyses.

## Results
### Overview and patient characteristics

We develop the Multi-omics Multi-cohort Assessment (MOMA) machine learning framework for predicting clinically actionable variations in cancer genomics, proteomics, and patient prognoses using histopathology images. Figure 1A and 1B show an overview of our interpretable machine learning methods. In brief, MOMA leverages robust image pre-processing (tiling, color normalization, and feature extraction), multiple-instance learning, and vision transformers to connect whole-slide pathology images with clinical and molecular profiles of interest. We further quantify the importance of each microenvironment component in each prediction task (Fig. 1C, D). To demonstrate the generalizability of our methods, we apply MOMA to multiple cohorts, including TCGA colorectal cancer cohorts (TCGA-COAD and TCGA-READ), the PLCO cohort, and the NHS and HPFS cohorts. Table 1 summarizes the demographic, molecular, and clinical profiles of patients in each cohort.

### MOMA predicts patients' overall survival and progression-free survival

Early-stage colorectal cancer patients have heterogeneous survival outcomes. Although many clinical and molecular predictors have been proposed, they cannot fully explain the divergent prognoses. To address this challenge, we employ MOMA to predict both overall survival and progression-free survival outcomes of stage I-II colorectal cancer patients. Results show that MOMA successfully identifies patients' overall survival outcomes in the TCGA held-out test set (Fig. 2A), with a concordance index (c-index) of 0.67 and log-rank test p-value of 0.01 between the two predicted prognostic groups. We further validate our model in two independent external cohorts: NHS-HPFS (Fig. 2B; *P* = 0.0495) and PLCO (Fig. 2C; *P* = 0.046), demonstrating the generalizability of our approaches. We visualize our models and show that dense clusters of adenocarcinoma cells are highly indicative of worse overall survival outcomes (Fig. 2D, E). Analyses that stratify colon cancer and rectal cancer samples show similar prediction performance in both cancer groups (Supplementary Data 1). Quantitative concept-based analyses reveal that regions of carcinoma cells, tumor-associated stroma, and interactions of carcinoma cells with smooth muscle cells in the cancerous regions are related to unfavorable overall survival (Fig. 1D).

In addition, MOMA reliably predicts the progression-free survival outcomes of the same cohorts of patients. In the TCGA held-out test set, our progression-free survival outcome prediction model achieves a c-index of 0.88 and a log-rank test *p*-value of 0.02 in distinguishing the prognostic groups (Fig. 3A). We further demonstrate the applicability of our model in the NHS-HPFS cohorts (Fig. 3B; c-index=0.6, *P* < 0.005). When stratifying the datasets into colon cancer and rectal

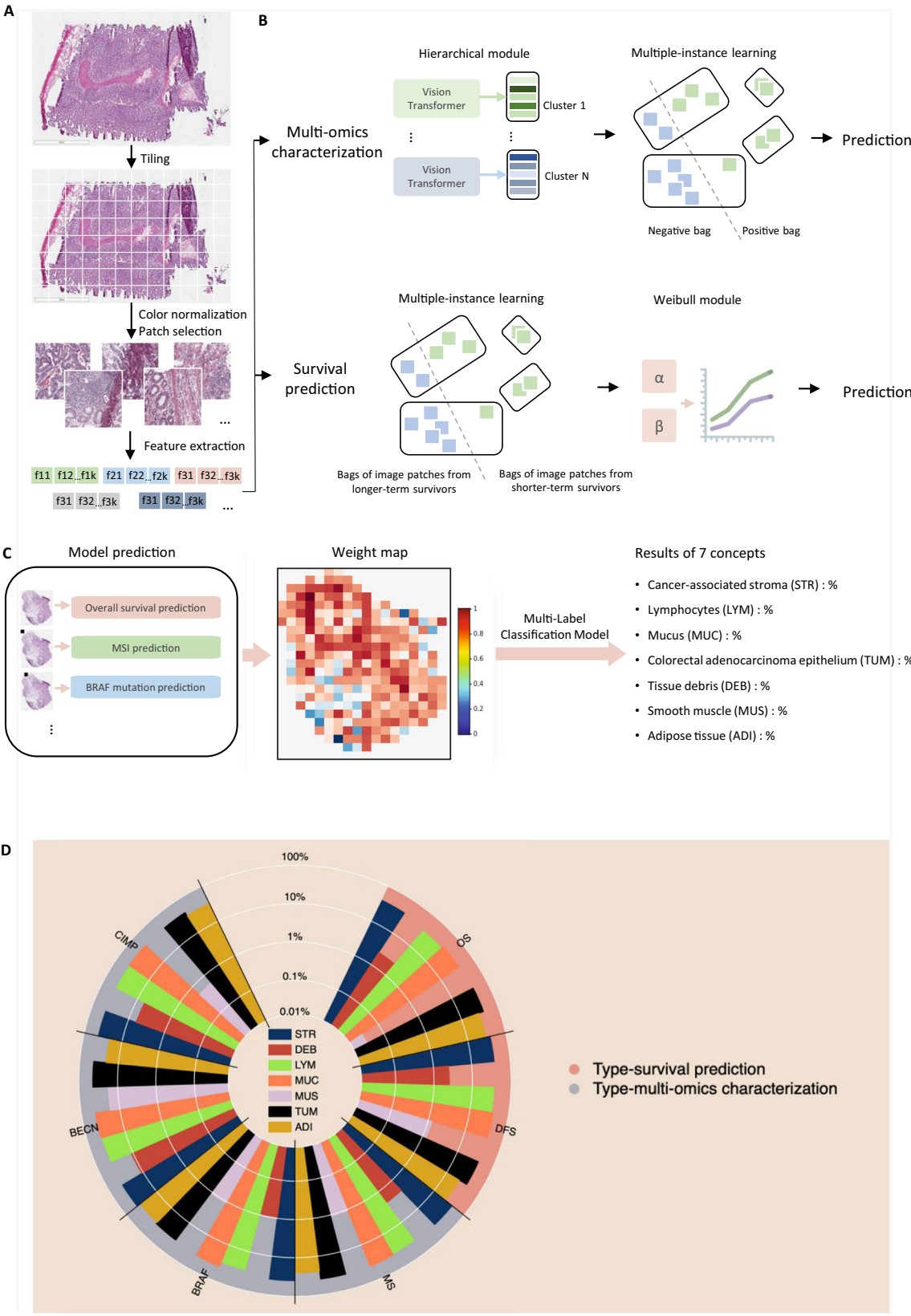

cancer groups, our approaches successfully identify the prognostic differences in both groups (Supplementary Data 1). A sensitivity analysis that was restricted to a surgery-only subgroup demonstrates the robustness of our results (Supplementary Fig. 1). Attention visualization shows that morphology patterns in tumor-associated stroma and groups of adenocarcinoma cells are highly indicative of progression-

free survival (Fig. 3C, D). Compared with the overall survival prediction, our progression-free survival model puts more emphasis on infiltrating lymphocytes and regions associated with extracellular mucin in its prediction.

Furthermore, we employ MOMA to predict both overall survival and progression-free survival outcomes of stage III colorectal cancer

**Fig. 1 | An overview of the Multi-omics Multi-cohort Assessment (MOMA) machine learning framework. A** Machine learning workflow for connecting whole-slide digital histopathology images with multi-omics biomarkers and survival outcomes. The MOMA platform processes the image patches from whole-slide pathology images, normalizes them, and leverages vision transformers to extract image features. **B** We develop multi-omics characterization and survival prediction frameworks using the extracted image features. **C** Model visualization and interpretation. To enhance the interpretability of our machine learning approaches, we compute the importance of each image region to the prediction target by quantifying the performance decay due to occlusion of the region, and we develop a multi-task classification model to quantify the concept (e.g., lymphocyte, stroma, tumor, adipose tissue, mucin, etc.) score using patches whose importance weight is greater than 0.7. This method connects prior histopathology knowledge with

quantitative importance metrics independently learned by the models. **D** A summary of the pathology concepts associated with survival and multi-omics predictions. The concept scores are plotted on the log scale. OS: overall survival prediction in early-stage CRC; DFS: disease-free survival prediction in early-stage CRC; MSI: microsatellite instability prediction; BRAF: *BRAF* mutation status prediction; BECN: BECN-1 overexpression prediction; CIMP: CpG island methylator phenotype prediction. The major concepts visualized here include lymphocytes (LYM), cancer-associated stroma (STR), tissue debris (DEB), mucus (MUC), smooth muscle (MUS), colorectal adenocarcinoma epithelium (TUM), and adipose tissue (ADI). The score for each concept indicates the relative importance of each type of microenvironment in predicting patient prognoses or the selected multi-omics variations with clinical implications.

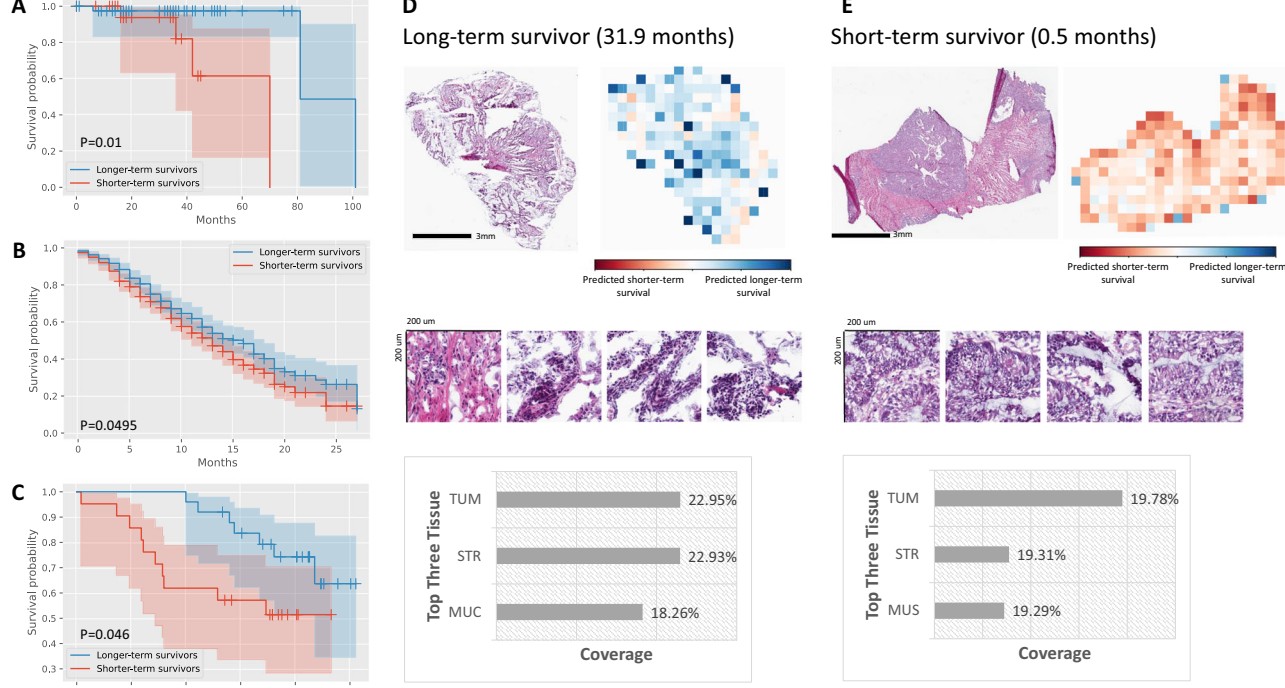

**Fig. 2 | MOMA predicts overall survival outcomes of stage I and II colorectal cancer patients using digital histopathology images, with validation in multiple independent cohorts. A** MOMA successfully distinguishes the shorter-term survivors from longer-term survivors using histopathology images (two-sided log-rank test *P*-value= 0.01). Results from the TCGA held-out test set are shown. **B** The machine learning model derived from MOMA is successfully validated in an independent external validation set from the Nurses' Health Study and Health Professionals Follow-up Study cohorts (two-sided log-rank test *P*-value<0.05).
**C** We further validate our overall survival prediction model in PLCO, a nationwide

multi-center study cohort (two-sided log-rank test *P*-value <0.05). **D** Model prediction of a patient with longer-term overall survival. The model focuses on regions of cancerous tissue and cancer-associated stroma when making the prediction in this example. **E** Interpretation of the overall survival prediction model. The prediction of a patient with shorter-term survival is shown in this figure panel. Cancerous tissue, cancer-associated stroma, and smooth muscle receive high attention weights in the overall survival prediction task. TUM: colorectal adenocarcinoma epithelium; STR: cancer-associated stroma; MUC: mucus; MUS: smooth muscle.

patients. Results show that MOMA successfully identifies patients' overall survival outcomes in the TCGA held-out test set (Fig. 4A), with a c-index of 0.66 and log-rank test p-value of 0.02 between the two predicted prognostic groups. We successfully validate our model in two independent external cohorts: NHS-HPFS (Fig. 4B; *P* = 0.0495) and PLCO (Fig. 4C; *P* = 0.04). On model visualization, we show that dense clusters of adenocarcinoma cells are highly indicative of worse overall survival outcomes (Fig. 4D, E). Similarly, MOMA successfully predicts patients' progression-free survival outcomes (Fig. 5A), with a c-index of 0.74 and log-rank test *p*-value of 0.02 between the two predicted prognostic groups in the TCGA held-out test set. These results are validated in our independent external cohorts from NHS-HPFS (Fig. 5B; *P* = 0.003). Similar to our overall survival results, model visualization shows that dense clusters of adenocarcinoma cells are highly indicative of worse progression-free outcomes (Fig. 5D, E). Quantitative concept-

based analyses reveal that regions of tumor-associated stroma and interactions of carcinoma cells with smooth muscle cells in the cancerous regions are related to unfavorable progression-free survival.

## MOMA provides improved prediction of MSI status using histopathology images

Immune checkpoint inhibitors have shown substantial survival benefits among a fraction of colorectal cancer patients. However, not all patients respond to this treatment modality with substantial immune-related adverse events. High-level microsatellite instability (MSI) status has been identified as a biomarker that predicts the response to immune checkpoint inhibitors. To facilitate the treatment effectiveness prediction for immune checkpoint inhibitors, we employ MOMA to predict the MSI status of each patient. Results show that the AUROC of the TCGA held-out test set is $0.88 \pm 0.06$ (Fig. 6A), and in the NHS-

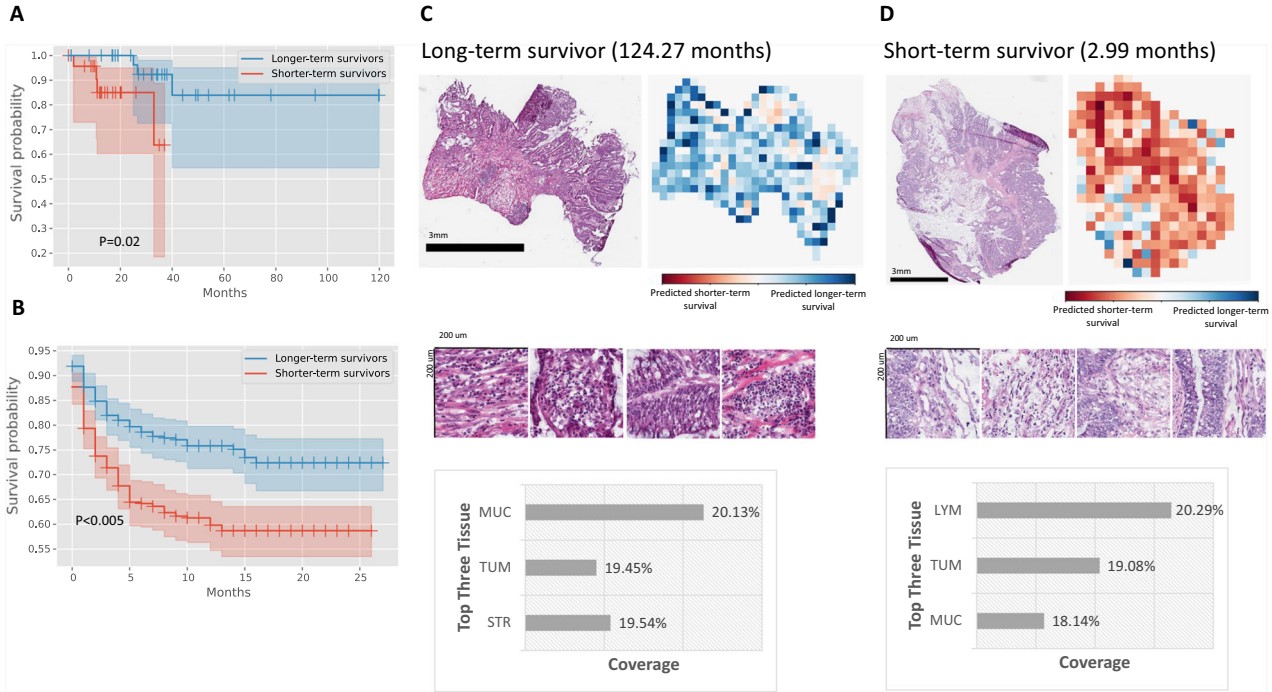

**Fig. 3 | Quantitative histopathology imaging predicts stage I and II colorectal cancer patients' progression-free survival outcomes. A** MOMA-trained models differentiate patients with early relapse or death from those with longer progression-free survival using histopathology images (two-sided log-rank test *P*-value=0.02). **B** We successfully validate our models using the independent external validation set from the Nurses' Health Study and Health Professionals Follow-up Study cohorts (two-sided log-rank test *P*-value<0.005). **C** Interpretation of the progression-free survival prediction model. The prediction of a patient with longer-term survival is shown in this figure panel. Mucosal regions and regions occupied by cancer cells both receive high attention weights in the overall survival prediction task. **D** Model prediction of a patient with shorter-term overall survival. In samples collected from shorter-term survivors, our model also focuses on regions of lymphocytes when making predictions. MUC: mucus; TUM: colorectal adenocarcinoma epithelium; STR: cancer-associated stroma; LYM: lymphocytes.

HPFS dataset, the AUROC is $0.76 \pm 0.04$ (Fig. 6B and Supplementary Table 1). Our methods improve the AUROC by 4% compared with the state-of-the-art methods by Kather et al.[12] (Supplementary Table 2). In both colon cancer and rectal cancer groups, MOMA shows correlations between histopathology images and MSI status (Supplementary Data 1). Model visualization further demonstrates that MOMA attends to lymphocytes, stroma, mucosa, and cancer regions when predicting MSI status (Fig. 6C, D).

### MOMA predicts copy number alterations (CNAs) and expression levels of key genes in cancer development

We further examine the performance of MOMA in predicting copy number alterations (CNAs), whole-genome doubling, and overexpression of the *BECN1* gene using histopathology images. CNAs of many key genes, including *FHIT* and *PTEN*, have been implicated in carcinogenesis. Here we show that MOMA predicts CNAs in *FHIT* and many other tumor suppressor genes (Fig. 7A–C). Compared with PC-CHiP, a commonly used image-based CNA prediction method, MOMA attains substantially improved prediction performance (Supplementary Table 3). In addition to the previously reported histopathology-CNA associations, MOMA further predicts amplifications in *NOL4L*, *HM13*, and *FOXS1*, and deletions in *WWOX* and *CCER1*, among many others (Fig. 7D–F). Furthermore, MOMA demonstrates improved prediction performance for whole-genome doubling, compared with PC-CHiP (Supplementary Table 4).

Moreover, MOMA reveals the correlation between histopathology image patterns and the expression levels of *BECN1* (Supplementary Fig. 2A), with the results validated in the NHS-HPFS dataset (Supplementary Fig. 2B and Supplementary Table 1). Stratified analyses by colon and rectal cancers show similar prediction performance in both cancer groups (Supplementary Data 1). In both *BECN1*-high and

*BECN1*-low tumors, the model focuses on tumor and mucus regions; however, in *BECN1*-high tumors, the model also focuses on regions occupied by lymphocytes, while in *BECN1*-low tumors the model focuses on the stroma. (Supplementary Fig. 2C and Supplementary Fig. 2D).

### MOMA identifies the histopathology patterns associated with BRAF mutation status

Genomic variations of proto-oncogenes and tumor suppressor genes are central to the development of colorectal cancers. For example, mutations in the *BRAF* gene propagate cell growth signals and are associated with reduced patient survival[23]. Several targeted therapy drugs focusing on *BRAF* inhibition have been developed, and combinatorial targeted therapy trials are underway. To identify the morphological impact of clinically important genomic variations, we leverage MOMA to systematically predict the mutation status of BRAF, HIF1A, and PIK3CA. Results show that MOMA identifies a moderate histopathology signal for predicting BRAF c.1799T > A (p.V600E) mutation in the TCGA test set, with an AUROC of $0.71 \pm 0.07$ (Supplementary Fig. 3A and Supplementary Table 1). To further identify the morphological patterns associated with this actionable genetic aberration, we visualize the attention distribution of our models in Supplementary Fig. 3B and Supplementary Fig. 3C. The concept scores of mucus, stroma, and tumor regions for BRAF mutation with c.1799T > A (p.V600E) detection are 19.89, 18.94, and 16.87, respectively (Fig. 1D). When classifying samples with BRAF mutation at any locus ($n = 529$) with those without BRAF mutation, we also show that MOMA can identify the morphological signals associated with BRAF mutations in general (Supplementary Fig. 4A, B). Similar approaches also identify the relationship between histopathology and the mutation status of *HIF1A* and *PIK3CA* (Supplementary Figs. 5 and 6).

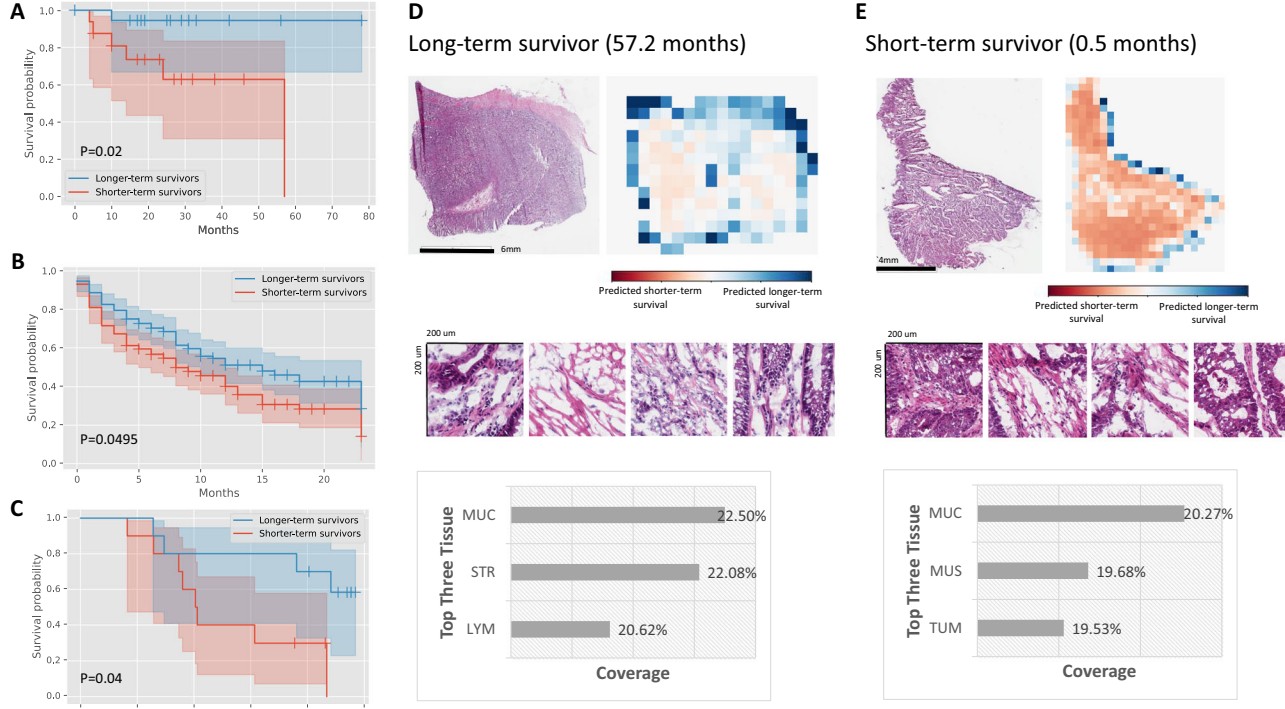

**Fig. 4 | MOMA predicts overall survival outcomes of stage III colorectal cancer patients using digital histopathology images, with validation in multiple independent cohorts. A** MOMA successfully distinguishes the shorter-term survivors from longer-term survivors using histopathology images (two-sided log-rank test *P*-value=0.02). Results from the TCGA held-out test set are shown. **B** The machine learning model derived from MOMA is successfully validated in an independent external validation set from the Nurses' Health Study and Health Professionals Follow-up Study cohorts (two-sided log-rank test *P*-value<0.05). **C** We further validate our overall survival prediction model in PLCO, a nationwide multi-center study cohort (two-sided log-rank test *P*-value = 0.04). **D** Model prediction of a patient with longer-term overall survival. The model focuses on regions of cancerous tissue and cancer-associated stroma when making the prediction in this example. **E** Interpretation of the overall survival prediction model. The prediction of a patient with shorter-term survival is shown in this figure panel. Cancerous tissue, cancer-associated stroma, and smooth muscle receive high attention weights in the overall survival prediction task. TUM: colorectal adenocarcinoma epithelium; STR: cancer-associated stroma; MUC: mucus; MUS: smooth muscle; LYM: lymphocytes.

## MOMA correlates histopathology patterns with the CpG island methylator phenotype

CpG island methylator phenotype (CIMP) colorectal cancer is a subtype characterized by widespread hypermethylation of promoter CpG islands. This hypermethylation pattern inactivates many tumor suppressor genes and causes global gene expression dysregulations and metabolic alterations. Previous studies suggest that patients with CIMP-high status have worse prognoses under the standard treatments[24]. To identify the histopathology patterns indicative of CIMP-high status, we employ MOMA to predict CIMP-high status and visualize the resulting model. Results show that the AUROC of the held-out test set in the TCGA cohort is 0.66 ± 0.06 (Supplementary Fig. 7A), and in the independent NHS-HPFS validation dataset, the AUROC is 0.63 ± 0.03 (Supplementary Fig. 7B and Supplementary Table 1). Furthermore, regions of lymphocytes and cancer cells are highly indicative of CIMP-high status (Supplementary Fig. 7C and Supplementary Fig. 7D).

## MOMA predicts consensus molecular subtypes using histopathology patterns

The consensus molecular subtype (CMS) is a commonly used molecular subtyping system for colorectal cancer that addresses inconsistencies in gene-expression-based classifications and reflects the biological differences in tumor characteristics[25]. To identify the histopathology patterns indicative of the CMS subtypes, we employ MOMA to classify the major CMS subtypes with sufficient numbers of samples (CMS2 and CMS4). Results show that MOMA achieved an AUROC of 0.66 ± 0.04 in the held-out test set not participating in the

model development process (Supplementary Fig. 8A and Supplementary Table 1). When stratifying the analysis by the colon and rectal cancer groups, we see a slightly improved performance in CMS prediction (AUROC = 0.74–0.77; Supplementary Data 1). MOMA indicates that regions of cancer-associated stroma and mucus are highly indicative of CMS2 and CMS4 (Supplementary Fig. 8B and Supplementary Fig. 8C).

## Comparisons of regions predictive of key clinical and multi-omics profiles

We summarize the regions indicative of patients' prognostic outcomes and molecular profiles identified by our interpretable machine learning framework. Our approaches provide a quantitative measurement of the relative importance of each region in predicting these outcomes of interest. For example, we show that histological patterns of the tumor, stroma, and mucus regions are relevant to the prediction of overall survival and disease-free survival, while regions with lymphocytes and mucus provide signals for predicting CIMP-high status. Figure 1D visualizes the regions of importance for each prediction task.

## Discussion

In this study, we designed the MOMA framework for molecular characterization and clinical prognostic prediction using histopathology images of colorectal cancer, and we further validated our models in two independent patient cohorts. Our results demonstrate that interpretable machine learning approaches can predict patients' survival outcomes and clinically important molecular profiles[26]. Our methods can automatically identify informative regions from whole-slide

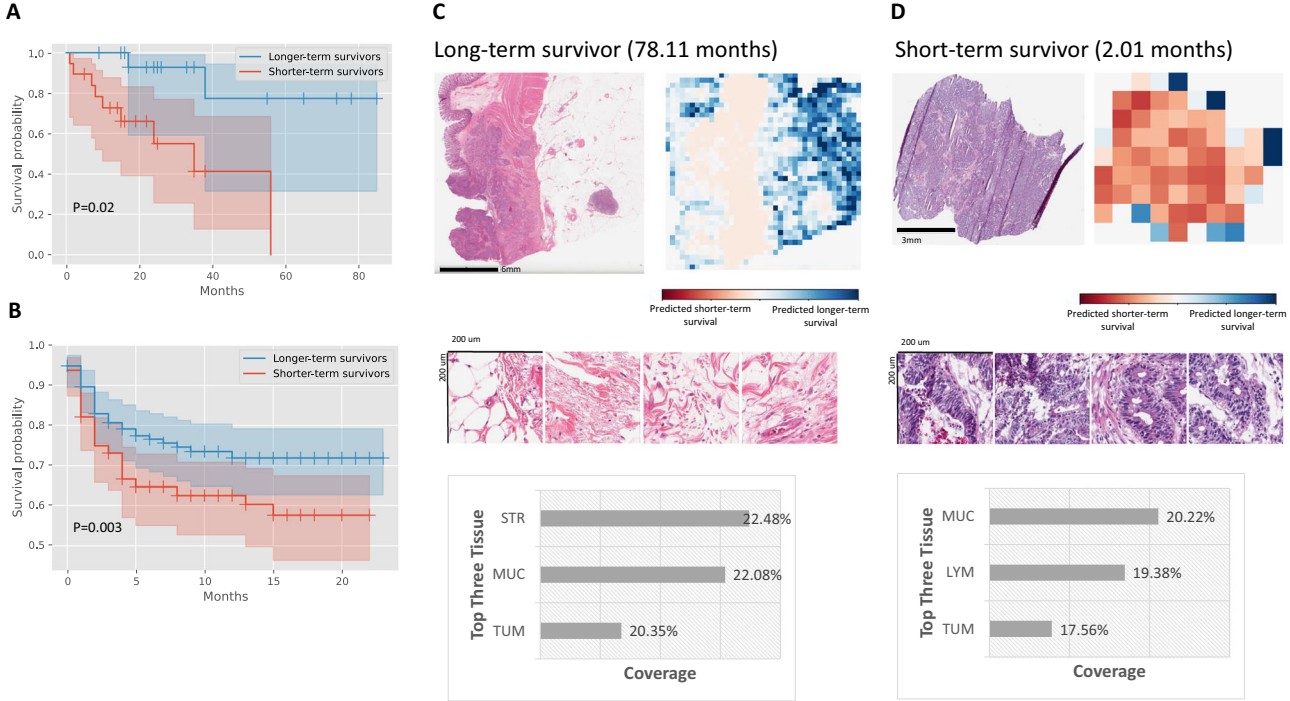

**Fig. 5 | MOMA predicts progression-free survival outcomes of stage III colorectal cancer patients using digital histopathology images, with validation in independent patient cohorts. A** MOMA successfully distinguishes the shorter-term survivors from longer-term survivors using histopathology images (two-sided log-rank test *P*-value=0.02). Results from the TCGA held-out test set are shown. **B** The machine learning model derived from MOMA is successfully validated in an independent external validation set from the Nurses' Health Study and Health Professionals Follow-up Study cohorts (two-sided log-rank test *P*-value=0.003). **C** Model prediction of a patient with longer-term progression-free survival. The model focuses on regions of cancerous tissue and cancer-associated stroma when making the prediction in this example. **D** Interpretation of the progression-free survival prediction model. The prediction of a patient with shorter-term survival is shown in this figure panel. Cancerous tissue, cancer-associated stroma, lymphocytes, and smooth muscle receive high attention weights in the overall survival prediction task. STR: cancer-associated stroma; MUC: mucus; TUM: colorectal adenocarcinoma epithelium; LYM: lymphocytes.

pathology images without the need for detailed region-level annotations. In addition, we employed the vision transformer and obtained significantly improved performance compared with that of standard deep learning methods[12]. Our multi-cohort validation showed the generalizability of our data-driven approaches for analyzing high-resolution digital pathology images.

Our models demonstrated that high-resolution histopathology slides contain useful predictive signals for genetic aberrations and survival outcomes. Because genetic profiling requires additional tissue samples, processing time, and costs, our prediction models that use only the H&E-stained histopathology slides can provide timely decision support for treatment selection in resource-limiting settings or in clinical scenarios with limited tissue availability. In addition, our stage-stratified survival outcome prediction successfully identified patients with shorter overall and disease-free survival under the standard treatments. These results showed that our machine learning approaches extracted stage-independent morphological signals indicative of patients' clinical outcomes. Because patient prognosis depends on many clinical factors, no prediction models can perfectly identify the survival outcomes of individual patients. Nonetheless, our approach unveiled histopathology patterns related to patient prognosis, which could be useful in guiding clinical decision-making. For example, clinicians may provide closer follow-up to patients with suboptimal clinical prognoses, consider more aggressive treatment options, or enroll them in ongoing clinical trials[27].

Compared with previously published methods, our approaches achieved substantially improved prediction performance. For instance, we first reproduced a widely used patch-based convolutional neural network[12] for MSI prediction using the TCGA dataset, and we showed that MOMA achieved a 4% improvement on the same dataset

(Supplementary Table 2). For CNA and WGD prediction, our approaches outperform models derived by the state-of-the-art PC-CHiP methods[28] by 7–29% (Fig. 7). Wilcoxon signed-rank tests confirmed that the performance difference is statistically significant in many clinically important genetic alterations, including *BCL2L1* amplification[29, 30] and *FHIT* deletion[31] (Supplementary Table 3). Furthermore, we successfully predicted the copy number alterations of 14 additional genes and connected our attention-based deep learning framework with time-to-event models for survival prediction. These methods have the potential to guide clinical decision-making, suggest clinical trial enrollment, and reduce costs attributed to sequencing by serving as a screening tool. We further validated our models in two independent patient populations, i.e., the NHS-HPFS and the PLCO cohorts, which demonstrated the reliability of our approaches when applied to previously unseen populations[32–35].

Our approaches provide several advantages compared with conventional methods. First, we embedded color normalization approaches in our end-to-end sample processing pipeline, which contributed to the improved robustness of our survival prediction models. In addition, we developed a tumor detection model trained on slide-level annotations and employed this model to identify the regions of interest for multi-omics and survival prediction tasks. Our approaches effectively reduced the need for detailed annotations by pathologists.

Due to the large number of parameters in deep learning models[36], they are largely viewed as black boxes with limited interpretability[37]. To enhance our understanding of model behaviors, we developed concept scores to quantitatively investigate the relevance of each region to the prediction tasks of interest, and we connected these regions with pathologists' annotations to provide biological insights into our data-driven models. Our results demonstrated that regions

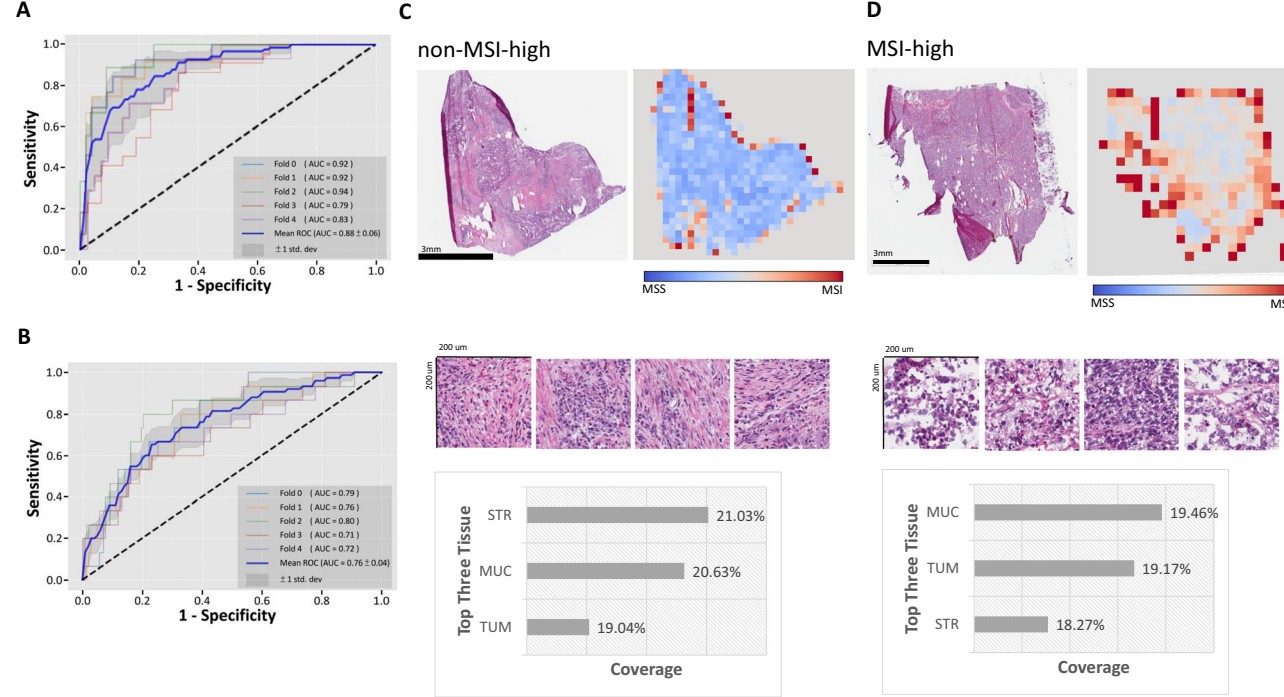

**Fig. 6 | MOMA predicts MSI status in colorectal cancer patients. A** Our MSI prediction model achieves an area under the receiver operating characteristic curve (AUC) of 0.88 in the TCGA held-out test set. **B** Our MSI prediction model is further validated in the Nurses' Health Study and Health Professionals Follow-up Study cohorts (AUC = 0.76). **C** Attention visualization of a pathology image with non-MSI-high cancer. Informative regions of cancer-associated stroma, cancer cells, and mucus in this prediction task are automatically highlighted by our trained machine learning model. **D** Attention visualization of a pathology image with MSI-high cancer. Regions with adenocarcinoma cells and their adjacent stroma receive high attention. STR cancer-associated stroma, MUC mucus, TUM colorectal adenocarcinoma epithelium.

occupied by tumors, supporting stroma, and mucus are crucial for survival prediction. These findings are consistent with observations that tumor invasiveness and tumor-stromal interactions are related to tumor progression[38]. We further revealed that lymphocyte-infiltrated regions are associated with MSI status, *BECN1* overexpression, and CpG island methylator phenotype (CIMP) status. The MSI-high status is an established biomarker for responses to immune checkpoint blockade[39], while *BECN1* is a key regulator of autophagy[40] and has been proposed as a potential target for immunotherapy[41]. Our findings confirmed the relevance of these biomarkers with immune cell infiltration and suggested the role of digital pathology profiling in the prediction of response to immunotherapy.

Our study has a few limitations. First, our study is based on patient populations in North America. Although our results are validated in two large-scale, independent, and diverse patient cohorts, additional studies that focus on specific patient populations are needed to evaluate the applicability of our models in the targeted clinical settings. In addition, recent studies on self-supervised machine learning hinted at the potential for enhanced representation learning for efficient machine learning[42-45], which may be useful for enhancing deep learning-based pathology feature extraction. Future research can investigate the benefits and potential caveats of these methods. Furthermore, incorporating patients' radiology imaging data, pathology profiles, molecular aberrations, and clinical characteristics may further improve the prognostic prediction for colorectal cancer patients. Additional research is required to identify the optimal prognostic prediction methods and enable personalized treatments and advance care planning.

In summary, we presented an interpretable machine-learning framework that systematically identifies the relationships between histopathology, molecular variations, and patients' survival outcomes. We successfully predicted key genetic aberrations, gene expression profiles, overall survival, and progression-free survival in colorectal cancer patients, with the results validated in two independent validation cohorts. Our approaches can be extended to characterize the prognostic-informing quantitative pathology patterns of other complex diseases.

## Methods

### Data sources

We obtained histopathology images of colon and rectal cancer patients from The Cancer Genome Atlas (TCGA) tissue slide dataset[46], the Prostate, Lung, Colorectal, and Ovarian Cancer Screening Trial (PLCO)[47], the Nurses' Health Study (NHS)[48], and the Health Professionals Follow-up Study (HPFS)[49,50]. We acquired the digital whole-slide pathology images, whole-exome sequencing results, and RNA-sequencing data of TCGA patients from the National Cancer Institute Genomic Data Commons Portal (https://portal.gdc.cancer.gov/). Mutation status, copy number alterations (including genetic amplifications and deletions), microsatellite instability, and CpG island methylator phenotypes (CIMP) of both colon and rectal adenocarcinomas were extracted from the cBioPortal (https://www.cbioportal.org/). Whole genome doublings and consensus molecular subtypes (CMS) of colorectal cancers were obtained from a previous TCGA publication[51].

In addition, we obtained PLCO data from the National Cancer Institute Cancer Data Access System, and we collected clinical, genomic profiles, immunohistochemistry, and hematoxylin-and-eosin (H&E) stained tissue microarray images from the NHS and the HPFS coordinated by Harvard T.H. Chan School of Public Health, Harvard Medical School, and Brigham and Women's Hospital. Notably, colorectal tumor tissue blocks in the NHS and the HPFS were retrieved from over a hundred hospitals throughout the U.S. with variable tissue age, which increased the generalizability of our

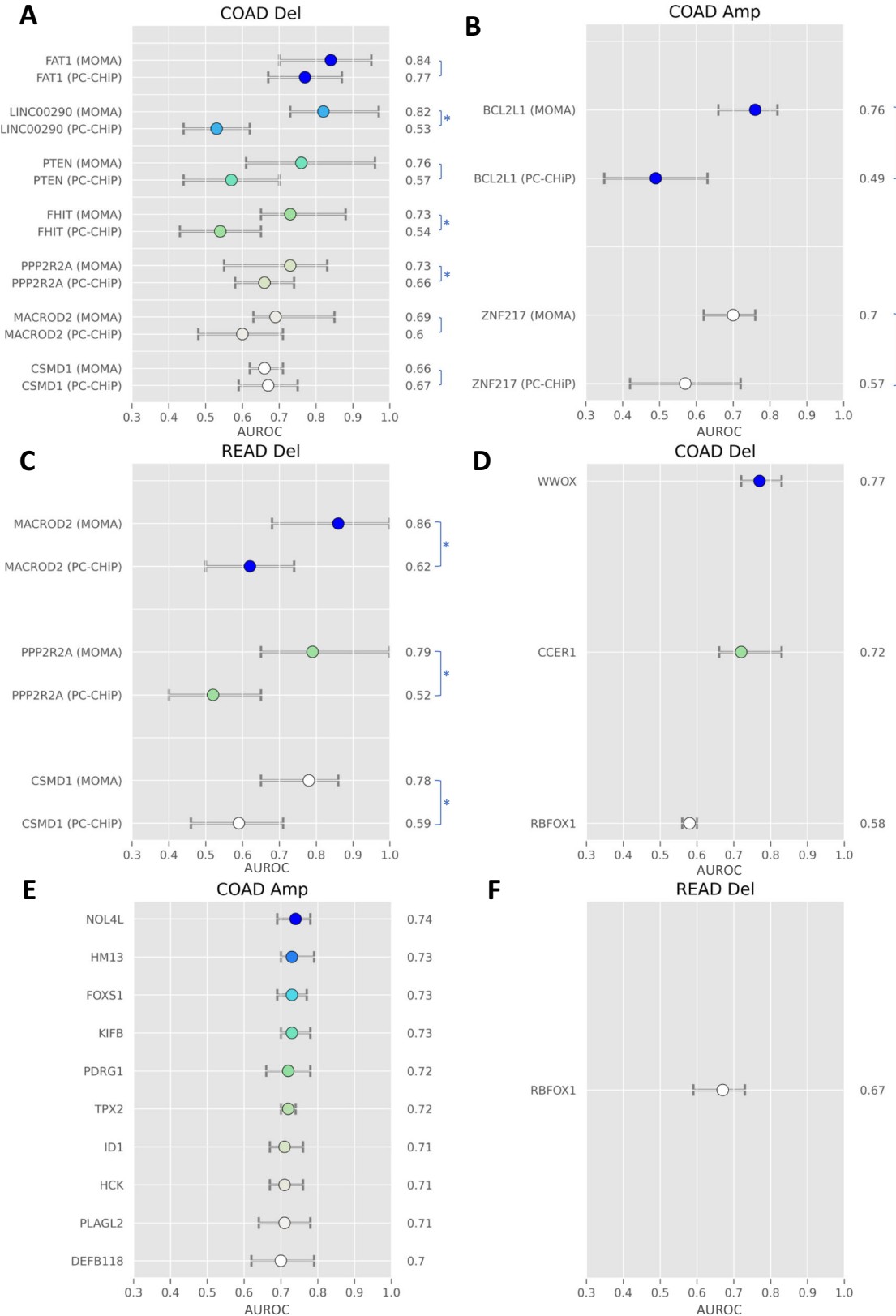

findings[52]. For each histopathology from the NHS and HPFS cohort, two experienced colorectal cancer pathologists reviewed the cancer samples and selected the cores to ensure the representativeness of the cores. Thus, the TMA images include regions of tumor cells, stroma, tumor/stromal interfaces (i.e., microscopic tumor invasive edges), lymphocyte infiltration, and other pathological changes

characteristic of the tumor sample from which the core was generated. Our multi-center study was approved by the Institutional Review Boards of Harvard Medical School (IRB20-1509). Our study protocol was also approved by the Brigham and Women's Hospital, Harvard T.H. Chan School of Public Health, and the participating registries as required.

**Fig. 7 | MOMA provides improved copy number alteration prediction compared with the current state-of-the-art methods and predicts additional copy number alterations not achieved in previous studies.** We systematically predict common copy number alterations of colorectal cancer tissues and compare the prediction performance with that of PC-CHiP[28]. The mean and range of AUROC are shown. **A** Prediction of common genetic deletions in patients with colon adenocarcinoma. **B** Prediction of common genetic amplification in patients with colon adenocarcinoma. **C** Prediction of common genetic deletions in patients with rectal adenocarcinoma. **D** Prediction of additional genetic deletions in colon adenocarcinoma. **E** Prediction of additional genetic amplifications in colon adenocarcinoma. **F** Prediction of additional genetic deletions in rectal adenocarcinoma. The error bars show the 95% confidence interval of the mean. In this analysis, 463 patients are in the COAD group, and 164 patients are in the READ group. Asterisks denote two-sided Wilcoxon signed-rank test P-value<0.05 when comparing the two groups.

## Overview of the Multi-omics Multi-cohort Assessment (MOMA) Framework

We designed a Multi-omics Multi-cohort Assessment (MOMA) machine learning framework to enable robust predictions of cancer genomics, proteomics, and important clinical outcomes. In this framework, we first pre-processed the whole-slide histopathology images by tiling each image into patches with 1000×1000 pixels, and we employed the color normalization proposed by Macenko et al.[53] to account for the staining differences across tissues and convert pixel values to a similar space in optical density. We used convolutional neural networks and vision transformers to extract pathology image features from each tile and connect these features with genomics, gene and protein expression levels, as well as patients' overall survival and disease-free survival outcomes.

One key feature of MOMA is the integration of multiple-instance learning[54], multi-modality outcome prediction frameworks[32], and biological interpretations of the prediction models. Leveraging state-of-the-art vision transformer models, we extracted informative features from the whole-slide images, and we connected them with genomic mutations, copy number alterations, transcriptomic profiles, and survival outcomes by incorporating relevant statistical models (e.g., Weibull models for survival prediction) to predict the molecular profiles and clinical outcomes of each individual. We further provided biological interpretations of the model predictions using pathology concepts of the tumor microenvironment[55]. Below we describe our methodology in detail.

## Multi-omics characterization via histopathology

Using the MOMA platform, we conducted multi-omics subgroup predictions on colorectal cancer patients, with a focus on clinically actionable molecular aberrations. Specifically, we predicted the microsatellite instability (MSI) status (65 MSI-high patients; 389 non-MSI-high patients in TCGA), CpG island methylator phenotype (CIMP; 58 CIMP-high tumor patients; 396 CIMP-low/negative tumor patients), *BRAF* c.1799T > A (p.V600E) mutation (48 patients with *BRAF* c.1799T > A (p.V600E) tumor; 529 patients with *BRAF* wild-type tumor), and the most prevalent Consensus Molecular Subtypes (CMS; 152 CMS2 (canonical) patients; 105 CMS4 (mesenchymal) patients) of colorectal cancer. These tasks were constructed as weakly-supervised classification tasks.

We employed the ResNet-50 network, a residual neural network with 50 layers[56], to extract 2048 features from the image patches. To mitigate the impact of artifacts in the whole-slide images, we applied the k-means algorithm to cluster the extracted feature vectors into 10 clusters, because typical colorectal cancer pathology images contain 10 different types of regions with biological significance (lymphocyte, stroma, debris, mucus, muscle, tumor, adipose tissue, background, normal, and others), and we used vision transformers[57] to derive the informative features for each cluster. The clusters whose weights were in the top three (strong positive association) or bottom three (strong negative association) would be used in the downstream analyses. After the vision transformer, the dimension of the feature vectors of each cluster was reduced to 512 to obtain efficient image representations for multiple-instance learning[58-60]. We used a trainable attention-based pooling operation to aggregate these feature vectors. Our transformer encoder layers contain 512

neurons in the hidden layer, 8 heads, and 2048 neurons in the multi-layer perceptron, with a dropout rate of 0.1.

Finally, we applied two loss functions in the prediction tasks. The first one was the bag loss function of standard binary or multiclass cross-entropy with the inverted class weights informed by the number of tiles in each class. The inverted class weights enabled machine learning models to account for the classes with fewer instances and prevent the models from biasing toward predicting all instances as the majority class. The other was the instance loss function of the tile-level classifiers. To compute the instance loss function, we first ranked the weights obtained from the attention-based multiple-instance learning to select the top three clusters with positive labels and the bottom three clusters with negative labels. Next, we employed the smooth support vector machine[61] with varying hyperparameter tau optimized for each task. We computed the total loss of the model as the sum of the bag loss function and the instance loss function.

To develop our models, we first split the TCGA dataset into 60% training, 20% validation, and 20% test sets. All tiles from the same whole-slide images were put in the same partition, in order to prevent information leaks. We trained our models using the training set, selected the optimal hyperparameters using the validation set, and reported our results in the untouched test set. We further validated our models in independent validation cohorts (please see the External Validation section). We reported the area under the receiver operating characteristic curves (AUROCs) of the test set for each classification task. We used a stochastic gradient descent (SGD) optimizer with a learning rate of 1e-3, a momentum of 0.9, a batch size of 1, and a weight decay of 5e-4. We trained all models with 250 epochs with a cosine annealing learning rate scheduler. We implemented our methods using Python 3.6 with PyTorch 1.6.0 in a single GPU system with NVIDIA Titan RTX. To make MOMA easily accessible to pathologists, oncologists, and biomedical informatics researchers, we further developed a web portal (https://rebrand.ly/MOMA_demo) that allows users to upload pathology images and employ our trained models to generate predictions. Our source codes for data analyses and trained models could be found at https://github.com/hms-dbmi/MOMA.

## Overall survival and progression-free survival prediction

To demonstrate the extensibility of our MOMA platform to different prediction tasks, we connected our machine learning framework with the Weibull modeling methods[62] to predict overall survival and progression-free survival outcomes of early-stage (stage I and stage II) and stage III colorectal cancer patients. We distinguished patients in the same stage groups into a "predicted longer-term survival group" and a "predicted shorter-term survival group," and we used the log-rank test to evaluate their differences in actual survival outcomes. Stage IV patients received heterogeneous treatments and were thus not included in our stratified survival outcome prediction analyses. The Weibull distribution is a probability distribution with shape (kappa) and scale (lambda) parameters. Combinations of the shape and scale parameters can model different hazard functions for survival analyses. We modified our machine learning framework to estimate these two statistical parameters in the Weibull survival model. We used a trainable attention-based pooling operation to aggregate the image feature vectors and employed the exponential activation function for lambda and softplus for kappa[63]. Our deep learning-based Weibull

modeling approach can handle right censoring and accommodate different patterns of death rate over time (e.g., increasing failure rate, decreasing failure rate, and constant failure rate). Due to the smaller sample size in the TCGA dataset for the survival prediction task (337 patients with stage I or II cancer and 181 stage III patients), we first conducted a 5-fold cross-validation on TCGA before validating our approaches in external validation cohorts. We divided the prediction results into shorter-term survival and longer-term survival groups using the median predicted survival index, and we tested the survival differences between the predicted groups using the log-rank test. We used an RMSprop optimizer with a learning rate of 1e-5 and trained the model with 5 epochs using a batch size of 1. We implement our training and testing processing using Python 3.6 with TensorFlow 2 in the same single GPU system.

## Multi-cohort external validation

To investigate the generalizability of our machine learning models, we harnessed two additional cohorts collected at different hospitals. Specifically, we used the pathology tissue microarray, genetic, immunohistochemistry, and clinical datasets from the NHS and HPFS to validate our trained prediction models. We further validated our survival prediction models using the whole-slide histopathology and survival information from the PLCO cohort. We applied the same image tiling, pre-processing, and color normalization methods to preprocess histopathology images from these external cohorts. We reported the AUROC (for classification tasks), concordance index (c-index), and log-rank test *p*-value (for survival prediction tasks) in these independent validations.

## Model visualization and interpretation

We further identified human-interpretable pathology features employed by our machine learning models to obtain biological insights into the connections between histopathology morphology and molecular profiles. We developed a model interpretation method that incorporates model-derived concept scores and expert-annotated concepts based on prior pathology knowledge. Specifically, we first quantified the importance of each image region by occluding all pixels in the region and computing the extent to which the predicted outcome changed when the region was occluded. We define the importance index of each image region as the numerical change of the predicted probability due to the occlusion of the region. To connect crucial regions with pathology interpretation, we leverage 100,000 histopathology images annotated by gastrointestinal pathologists with seven concepts: colorectal adenocarcinoma epithelium, cancer-associated stroma, lymphocytes, smooth muscle, mucus, adipose tissue, and tissue debris. We developed a deep learning model that classified image regions into these pathology concepts with an accuracy of 99.38%, and we employed this model to compute the concept scores for regions with importance indices greater than 0.7. We scaled our concept scores to a range of [0, 100], where 100 indicates the region has the highest relevance to the concept of interest. Thus, our concept scores indicate the amount of attention our machine learning model pays to different regions of pathology changes in making the prediction, and it is not directly related to the amount of area occupied by each pathology pattern in the slides. We repeated this process for each machine learning task we performed. We used the standard color map to visualize the importance index and overlaid it with the original histopathology images. These approaches provide intuitive model interpretations using well-established concepts in cancer pathology.

## Statistics & reproducibility

No data were excluded from the analyses. All available samples were included in the machine learning analyses, and the experiments were not randomized. The investigators were blinded to the labels of the samples in the test set before the final model evaluation.

## Reporting summary

Further information on research design is available in the Nature Portfolio Reporting Summary linked to this article.

## Data availability

TCGA histopathology, molecular, and clinical data used in this study are available through the Genomic Data Commons portal [https://portal.gdc.cancer.gov/]. Mutation status, copy number alterations (including genetic amplifications and deletions), microsatellite instability, and CpG island methylator phenotypes (CIMP) of TCGA samples were extracted from the cBioPortal [https://www.cbioportal.org/]. The PLCO data is available at the National Cancer Institute Cancer Data Access System [https://cdas.cancer.gov/plco/]. The data from Nurses' Health Studies and the Health Professionals Follow-up Study are available under restricted access due to patient privacy considerations. Procedures to access the data are described at https://www.nurseshealthstudy.org/researchers (contact email: nhsaccess@channing.harvard.edu) and https://sites.sph.harvard.edu/hpfs/forcollaborators/.

## Code availability

The codes for our data analyses and trained models could be found at https://github.com/hms-dbmi/MOMA. The demo website of our MOMA system is at https://rebrand.ly/MOMA_demo.

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

## Acknowledgements

We thank Li-Jin Huang for her exploratory data analyses, Shih-Yen Lin for his suggestions on manuscript revision, and Shannon Gallagher for her administrative support. K-H.Y. is partly supported by the National Institute of General Medical Sciences grant R35GM142879, Google Research Scholar Award, and the Blavatnik Center for Computational Biomedicine Award. We thank the AWS Cloud Credits for Research, Microsoft Azure for Research Award, the NVIDIA GPU Grant Program, and the Extreme Science and Engineering Discovery Environment (XSEDE) at the Pittsburgh Supercomputing Center (allocation TG-BCS180016) for their computational support. The Nurses' Health Study and the Health Professionals Follow-up Study were supported in part by U.S. National Institutes of Health (NIH) grants (P01 CA87969, UM1 CA186107, P01 CA55075, UM1 CA167552, U01 CA167552, R35 CA197735, R01 CA151993, R01 CA248857); by Cancer Research UK Grand Challenge Award (UK C10674/A27140 to the OPTIMISTICC Team). Funds for this work were provided to P.-C.T., T.-H.L., K.-C.K., F.-Y.S., J-H.C. by the National Science and Technology Council (NSTC), Taiwan (MOST 110-2634-F-006-021 and NSTC 111-2634-F-006-011) and National Center for High-performance Computing (NCHC), Taiwan. We would like to thank the participants and staff of the Nurses' Health Study and the Health Professionals Follow-up Study for their valuable contributions as well as the following state cancer registries for their help: AL, AZ, AR, CA, CO, CT, DE, FL, GA, ID, IL, IN, IA, KY, LA, ME, MD, MA, MI, NE, NH, NJ, NY, NC, ND, OH, OK, OR, PA, RI, SC, TN, TX, VA, WA, WY. The authors assume full responsibility for the analyses and interpretation of these data.

## Author contributions

P.-C.T., T-H.L., K-C.K. performed the analyses and wrote the manuscript, F-Y.S. and E.M. interpreted the results and edited the manuscript, T.U., M.Z., M.C.L., J.P.V., M.G., Y.T., S.M.K., K.W., M.S., J.A.M., and A.T.C., contributed to the collection of data from the Nurses' Health Study and the Health Professionals Follow-up Study and edited the manuscript. J.H.C. designed the study, supervised the work, and edited the manuscript, J.N. and S.O. contributed to the collection of data from the Nurses' Health Study and the Health Professionals Follow-up Study, interpreted the results, and edited the manuscript. K-H.Y. conceived and designed the study and analyses, obtained the data, interpreted the results, supervised the work, and wrote and revised the manuscript.

## Competing interests

K-H.Y. is an inventor of US 16/179,101, entitled "Quantitative Pathology Analysis and Diagnosis using Neural Networks." This patent is assigned to Harvard University. K-H.Y. was a consultant of Curatio. DL. K.W. is currently a stakeholder and employee of Vertex Pharmaceuticals. This study was not funded by this entity. All other authors have nothing to disclose.
