## [Peer Review File · Nature Communications]

This manuscript has been previously reviewed at another journal that is not operating a transparent peer review scheme. This document only contains reviewer comments and rebuttal letters for versions considered at *Nature Communications*.

REVIEWER COMMENTS

Reviewer #1 (Remarks to the Author): clinical expertise in colorectal cancer

Comments

It is known that the colorectal cancer is one of the earliest cancer types whose genomic feature have been dissected. AI-driven exploration of the association between Histopathology Images and the molecular feature of CRC could have great significance for the clinical diagnosis of this deadly disease. In the current study, Tsai and colleagues developed a novel explainable machine learning method MOMA to predict molecular alteration and clinical profile of CRC with histopathology image data. Specifically, MOMA integrates multiple prediction frameworks as a whole, which is the key feature shown in this study. Overall, this study is close to the clinic and might be applicable in the future if the performance could be improved and validated in more datasets with different ethnic composition. In addition to the topic and significance, I have the following concerns that need to be addressed.

My Concerns:

1. CMS(consensus molecular sub) system is one of the most popular molecular subtyping system for colorectal cancer, which linked to many genomic features like MSI, CIMP and altered signaling pathways. Could MOMA be used for predicting CMS subtypes? I think It would be more informative if this model were made.
2. Unbalanced positive and negative dataset used in training dataset(MSI: 65 vs389; CIMP: 58 vs 396; BRAF: 36vs524), and the bias maybe they split the dataset into the training, validation and test dataset by 3:1:1. Therefore, this bias should be noted during the model training process, or at least discussed?
3. Why Only early-stage CRCs(I-II) were used for classifying short and long survivors and PFS. The author stated that "MOMA can be generalizable to multiple patient cohorts", I think that the stage III and IV shall not be ignored. Also, in general, early-stage CRC has better survival than stage III, IV CRC, so I think the definition of "short" and "long" survivor (limited in early-stage CRC) is not precise here.
4. The performance of MOMA is a little confusing. Although the authors described some texts on the performance of MOMA when compared with previous approaches, no detail data were shown. e.g., the 3rd point of "Result" section entitles "MOMA provided improved prediction of MSI status", while only the performance score of MOMA was displayed.
5. Clinical value of MOMA is insufficiently evaluated. As the MOMA is designed for multiple tasks on clinical practice such as survival and alteration prediction, no data show the clinical value, i.e. can it substitute the current assays used by clinical platforms? For example, when MOMA is applied to predict the CRC progression (i.e. OS or PFS), how it compares to the known clinical features such staging,
6. The authors sometimes describe MOMA as a platform in the manuscript, however, I cannot find a web service or software/portal for potential MOMA users based on code availability.
7. The 2nd paragraph of introduction lacks the description about related research advances in the CRCs.
8. Figure 1C/D and related descriptions are not clear, what can readers get from the quantification of microenvironment component?
9. It is weird that the author specifically highlight the "BECN1" in the CNV prediction section, please clarify.

Reviewer #2 (Remarks to the Author): expertise in machine learning and cancer prediction

In this paper, the authors developed a multi-omics multi-cohort assessment (MOMA) platform based on digital pathology images to predict molecular biomarkers and patient survival in colorectal cancer. Strengths of the paper include:

1. Three large patient cohorts (n=1,888)
2. Methodology is state-of-the-art and clearly described.
3. Model visualization and interpretation are a plus

Weaknesses include:

1. It is uncertain if proposed framework has any use clinically. First, prediction of patient prognosis

may not make any difference in patient management. Second, the performance on using digital pathology images to predict survival and molecular biomarkers is only moderate. For example, c-index for predicting survival is only 0.67. The AUROC for predicting BRAF is only 0.68 in the TCGA test set and 0.66 in the NHS-HPFS cohort. The AUROC for predicting CIMP is only 0.65 for TCGA and 0.64 for the NHS-HPFS. The proposed use for the model is to "provide timely decision support for treatment selection in resource-limiting regions using only the H&E-stained histopathology slides or in settings of limited tissue availability". However, with the current performance, it is uncertain if this is possible. Combination with other modalities such as imaging or clinical variables may be able to achieve better results.

2. Reporting the AUROCs alone is not enough for machine learning study to predict binary outcomes. Other performance metrics include accuracy, sensitivity and specificity. For imbalanced outcomes as studied in this paper, precision-recall is a very helpful performance metric to add.

3. No detail is given on how the genomic data are extracted from the data source (e.g., from the TCGA).

4. Colon and rectal cancer are sufficiently different to warrant subgroup analyses rather than grouping together as one cohort.

5. In the prediction of outcome, only stage I and stage II colorectal cancer is used. However, the treatment and prognosis for colon and rectal cancer are very different as well as the work-up. For example, the first-line diagnostic imaging for rectal cancer is MRI and it is frequently treated with radiation before surgery.

6. It is unclear throughout the manuscript what state-of-the-art technique is MOMA being compared to and what the results are.

Reviewer #3 (Remarks to the Author): expert in colorectal cancer pathology

This is a well-designed study evaluating the MOMA platform in colorectal cancer patients. The MOMA platform combines histological and genomic data (copy number alterations, gene expression profile and microsatellite instability status). The authors have validated the platform in a large cohort and some of the results are expected or well known (eg. lymphocyte rich population at the invasive margin of the tumour associated with MSI)

The results are quite robust, and this is a very welcome paper in the development of AI in histopathology. However there are several concerns/ limitations in this paper that need to be addressed

1. The study includes stage I and stage II CRC but, particularly stage II, is a very heterogeneous group in which risk factors (iv. vascular invasion -mainly extramural-, tumour budding...) play an important role in the prognosis. There is no mention in the paper that these parameters were evaluated. Depending on the presence of some of this risk factors stage II CRC patients might receive chemoradiotherapy and the prognosis (DFS, OS...) will be different in this subgroup. Is there any data on treatment (were all the patients chemo-naïve or some patients received adjuvant chemotherapy)

2. Contrary to the first set of patients /samples studied, the validation cohort included TMAs. This might have affected the results. The authors should explain how they assessed the 10 types of regions in the TMAs. Did TMAs include different areas from the tumour (eg. Core, stroma, invasive front)? or only 1 core per tumour was analysed or only one region per tumour?

3. The conclusion on "dense clusters of adenocarcinoma cells being highly indicative of worse OS" (page 13) contradicts current knowledge that tumour with low-stroma (ie high proportion of epithelial cells and low amount of stroma) behave better than tumour with high-stroma content (GW van Pelt 2018; J Gao 2021...). This needs clarification

4. The entire discussion around BRAF and response to double/ triple chemotherapy (BEACON study) needs to be properly addressed. The authors claim that MOMA can predict BRAF mutation but not what type of mutation. The only actionable mutation is V600E but around 20% of CRC patients have non-V600E mutations (of uncertain clinical significance but clearly not associated

with poor prognosis or response to encorafenib-cetuximab-binimetinib). Without knowledge of what type of BRAF mutation the data is of very limited value.

RESPONSE TO REVIEWERS' COMMENTS

Reviewer #1 (Remarks to the Author): clinical expertise in colorectal cancer

Comments

It is known that the colorectal cancer is one of the earliest cancer types whose genomic feature have been dissected. AI-driven exploration of the association between Histopathology Images and the molecular feature of CRC could have great significance for the clinical diagnosis of this deadly disease. In the current study, Tsai and colleagues developed a novel explainable machine learning method MOMA to predict molecular alteration and clinical profile of CRC with histopathology image data. Specifically, MOMA integrates multiple prediction frameworks as a whole, which is the key feature shown in this study. Overall, this study is close to the clinic and might be applicable in the future if the performance could be improved and validated in more datasets with different ethnic composition. In addition to the topic and significance, I have the following concerns that need to be addressed.

Our response: We thank the reviewer for appreciating the novelty, scientific significance, and clinical relevance of our machine learning-based Multi-Omics, Multicohort Assessment (MOMA) of digital pathology images. Below we answer the reviewer's question in detail.

My Concerns:

1. CMS(consensus molecular sub) system is one of the most popular molecular subtyping system for colorectal cancer, which linked to many genomic features like MSI, CIMP and altered signaling pathways. Could MOMA be used for predicting CMS subtypes? I think It would be more informative if this model were made.

Our response: We thank the reviewer for the suggestions. Following the reviewer's suggestion, we have added a new analysis that uses MOMA to predict CMS subtypes. Our new results show that MOMA predicted CMS subtypes with an AUROC of 0.74 for colon cancer, an AUROC of 0.77 for rectal cancer, and AUROC = 0.66 when training a single model for both colon and rectal cancers (the new Supplemental Table 2 and Supplemental Figure 8). Although the prediction performance is not sufficiently high to warrant replacing molecular profiling for CMS classification, our results reveal that hematoxylin-and-eosin (H&E)-stained histopathology images contain previously unrecognized biological signals related to CMS subtypes. We have included these new results in the revised manuscript.

In the revised Methods section (page 8, the second paragraph of the revised manuscript):

Multi-omics Characterization via Histopathology

Using the MOMA platform, we conducted multi-omics subgroup predictions on colorectal cancer patients, with a focus on clinically actionable molecular aberrations. Specifically, we predicted the microsatellite instability (MSI) status (65 MSI-high patients; 389 non-MSI-high patients in TCGA), CpG island methylator phenotype (CIMP; 58 CIMP-high tumor patients; 396 CIMP-low/negative tumor patients), *BRAF* c.1799T>A (p.V600E) mutation (48 patients with *BRAF* c.1799T>A (p.V600E) tumor; 529 patients with *BRAF*-wild-type tumor), and the most prevalent Consensus Molecular Subtypes (CMS; 152 CMS2 (canonical) patients; 105 CMS4 (mesenchymal) patients) of colorectal cancer. These tasks were constructed as weakly-supervised classification tasks.

In the revised Results section (page 19, the second paragraph):

MOMA Predicted Consensus Molecular Subtypes using Histopathology Patterns

The consensus molecular subtype (CMS) is a commonly used molecular subtyping system for colorectal cancer that addresses inconsistencies in gene-expression-based classifications and reflects the biological differences in tumor characteristics⁴². To identify the histopathology patterns indicative of the CMS subtypes, we employed MOMA to classify the major CMS subtypes with sufficient numbers of samples (CMS2 and CMS4). Results show that MOMA achieved an AUROC of 0.66 ± 0.04 in the held-out test set not participating in the model development process (Supplemental Figure 8A and Supplemental Table 1). When stratifying the analysis by the colon and rectal cancer groups, we saw a slightly improved performance in CMS prediction (AUROC = 0.74-0.77; Supplemental Table 2). MOMA indicated that regions of cancer-associated stroma and mucus are highly indicative of CMS2 and CMS4 (Supplemental Figure 8B and Supplemental Figure 8C). In particular, the concept scores of lymphocytes and tumors are 21.60% and 19.46%, respectively.

The new Supplemental Figure 8: MOMA identified the association between CMS and histopathology image patterns. (A) MOMA characterized a moderate correlation between CMS2 and CMS4 in histopathology image features. Results from the TCGA held-out test set were shown. (B) Attention visualization of a histopathology image from a CMS2 patient. (C) Attention

visualization of a histopathology image from a CMS4 patient. Regions of stroma, cancers, lymphocytes, and mucus received high attention in this molecular classification task. MUC: mucus; TUM: colorectal adenocarcinoma epithelium; STR: cancer-associated stroma; LYM: lymphocytes.

2. Unbalanced positive and negative dataset used in training dataset(MSI: 65 vs389; CIMP: 58 vs 396; BRAF: 36vs524), and the bias maybe they split the dataset into the training, validation and test dataset by 3:1:1. Therefore, this bias should be noted during the model training process, or at least discussed?

Our response: We thank the reviewer for this comment. Indeed, unbalanced classes pose challenges to training machine learning models. We employed all samples with available molecular profiles and randomly divided them into training, validation, and test datasets. Thus, the unbalanced classes reflect the prevalence of each class in our cohort. To address the challenge of unbalanced classes in the development of prediction models, we employed bag loss functions with inverted class weights informed by the number of instances in each class. Our approach allows machine learning models to account for the classes with fewer instances and prevent the model from biasing toward predicting all instances as the majority class. And we successfully validated our machine learning models in the independent Nurses' Health Study and Health Professionals Follow-up Study (NHS-HPFS) datasets. We have revised the Methods section to clarify this point.

In the revised Methods section (page 9, the second paragraph):

Finally, we applied two loss functions in the prediction tasks. The first one was the bag loss function of standard binary or multiclass cross-entropy with the inverted class weights informed by the number of tiles in each class. The inverted class weights enabled machine learning models to account for the classes with fewer instances and prevent the models from biasing toward predicting all instances as the majority class. The other was the instance loss function of the tile-level classifiers. To compute the instance loss function, we first ranked the weights obtained from the attention-based multi-instance learning to select the top three clusters with positive labels and the bottom three clusters with negative labels. Next, we employed the smooth support vector machine³⁷ with varying hyperparameter tau optimized for each task. We computed the total loss of the model as the sum of the bag loss function and the instance loss function.

3. Why Only early-stage CRCs(I-II) were used for classifying short and long survivors and PFS. The author stated that "MOMA can be generalizable to multiple patient cohorts", I think that the stage III and IV shall not be ignored. Also, in general, early-stage CRC has better survival than stage III, IV CRC, so I think the definition of "short" and "long" survivor (limited in early-stage CRC) is not precise here.

Our response: We thank the reviewer for this suggestion. In the revised manuscript, we have added a new analysis that predicts the overall survival and progression-free survival outcomes of stage III patients (the new Figures 4 and 5). Our results show that MOMA can distinguish shorter-term survivors from longer-term survivors among stage III patients receiving standard treatments as well. We further validated our results in independent patient cohorts with sufficient sample sizes. Because stage IV patients in our cohorts received heterogeneous treatments and we have a smaller sample size, we did not apply our approaches to stage IV patients.

In addition, we have clarified the definition of "shorter-term" and "longer-term" survivors among stage I and stage II patients to highlight the additional prognosis-informing signals from whole-slide pathology images independent of the clinical stage. Taken together, we have successfully predicted the survival outcomes of both early-stage (stages I and II) and stage III CRC patients. We have revised our manuscript to reflect these changes.

In the revised Methods section (page 10, the second paragraph):

Overall Survival and Progression-Free Survival Prediction

To demonstrate the extensibility of our MOMA platform to different prediction tasks, we connected our machine learning framework with the Weibull modeling methods³⁸ to predict overall survival and progression-free survival outcomes of early-stage (stage I and stage II) and stage III colorectal cancer patients. We distinguished patients in the same stage groups into a “predicted longer-term survival group” and a “predicted shorter-term survival group,” and we used the log-rank test to evaluate their differences in actual survival outcomes. Stage IV patients received heterogeneous treatments and were thus not included in our stratified survival outcome prediction analyses.

In the revised Results section (page 15, the second paragraph):

Furthermore, we employed MOMA to predict both overall survival and progression-free survival outcomes of stage III colorectal cancer patients. Results showed that MOMA successfully identified patients’ overall survival outcomes in the TCGA held-out test set (Figure 4A), with a c-index of 0.66 and log-rank test p-value of 0.02 between the two predicted prognostic groups. We successfully validated our model in two independent external cohorts: NHS-HPFS (Figure 4B; P = 0.02) and PLCO (Figure 4C; P = 0.02). On model visualization, we showed that dense clusters of adenocarcinoma cells are highly indicative of worse overall survival outcomes (Figures 4D and 4E). Similarly, MOMA successfully predicted patients’ progression-free survival outcomes (Figure 5A), with a c-index of 0.74 and log-rank test p-value of 0.02 between the two predicted prognostic groups in the TCGA held-out test set. These results are validated in our independent external cohorts from NHS-HPFS (Figure 5B; P = 0.003). Similar to our overall survival results, model visualization showed that dense clusters of adenocarcinoma cells are highly indicative of worse progression-free outcomes (Figures 5D and 5E). Quantitative concept-based analyses revealed that regions of tumor-associated stroma and interactions of carcinoma cells with smooth muscle cells in the cancerous regions were related to unfavorable progression-free survival.

The new Figure 4. MOMA predicted overall survival outcomes of stage III colorectal cancer patients using digital histopathology images, with the results validated in multiple independent cohorts. (A) MOMA successfully distinguished the shorter-term survivors from longer-term survivors using histopathology images (log-rank test P-value = 0.02). Results from the TCGA

held-out test set are shown. (B) The machine learning model derived from MOMA is successfully validated in an independent external validation set from the Nurses' Health Study and Health Professionals Follow-up Study cohorts (log-rank test P-value < 0.05). (C) We further validated our overall survival prediction model in PLCO, a nationwide multi-center study cohort (log-rank test P-value = 0.04). (D) Model prediction of a patient with longer-term overall survival. The model focused on regions of cancerous tissue and cancer-associated stroma when making the prediction in this example. (E) Interpretation of the overall survival prediction model. The prediction of a patient with shorter-term survival is shown in this figure panel. Cancerous tissue, cancer-associated stroma, and smooth muscle receive high attention weights in the overall survival prediction task. TUM: colorectal adenocarcinoma epithelium; STR: cancer-associated stroma; MUC: mucus; MUS: smooth muscle; LYM: lymphocytes.

The new Figure 5. MOMA predicted progression-free survival outcomes of stage III colorectal cancer patients using digital histopathology images, with the results validated in independent patient cohorts. (A) MOMA successfully distinguished the shorter-term survivors from longer-term survivors using histopathology images (log-rank test P-value = 0.02). Results from the TCGA held-out test set are shown. (B) The machine learning model derived from MOMA is successfully validated in an independent external validation set from the Nurses' Health Study and Health Professionals Follow-up Study cohorts (log-rank test P-value = 0.003). (C) Model prediction of a patient with longer-term progression-free survival. The model focused on regions of cancerous tissue and

cancer-associated stroma when making the prediction in this example. (D) Interpretation of the progression-free survival prediction model. The prediction of a patient with shorter-term survival is shown in this figure panel. Cancerous tissue, cancer-associated stroma, lymphocytes, and smooth muscle receive high attention weights in the overall survival prediction task. STR: cancer-associated stroma; MUC: mucus; TUM: colorectal adenocarcinoma epithelium; LYM: lymphocytes.

4. The performance of MOMA is a little confusing. Although the authors described some texts on the performance of MOMA when compared with previous approaches, no detail data were shown. e.g., the 3rd point of "Result" section entitles "MOMA provided improved prediction of MSI status", while only the performance score of MOMA was displayed.

Our response: We thank the reviewer for this question. Following the reviewer's suggestion, we have included supplemental tables (the revised Supplemental Tables 3, 4, and 5) to compare the prediction performance of MOMA with the state-of-the-art models for predicting MSI, copy number alterations (CNAs), and whole-genome doubling. In the revised manuscript, we have conducted a Wilcoxon rank sum test to illustrate the performance difference and show that MOMA has significantly better performance in predicting CNAs of key genes in colon and rectal cancers, as well as in predicting whole-genome doubling in

rectal cancer, compared with previous methods. We have included these results in the revised Results section.

In the revised Results section (page 16, the first paragraph):

MOMA Provided Improved Prediction of MSI Status using Histopathology Images

To facilitate the treatment effectiveness prediction for immune checkpoint inhibitors, we employed MOMA to predict the MSI status of each patient. Results showed that the AUROC of the TCGA held-out test set is 0.88 ± 0.06 (Figure 6A), and in the NHS-HPFS dataset the AUROC is 0.76 ± 0.04 (Figure 6B and Supplemental Table 1). Our methods improved the AUROC by 4% compared with the state-of-the-art methods¹³ (Supplemental Table 3). In both colon cancer and rectal cancer groups, MOMA showed correlations between histopathology images and MSI status (Supplemental Table 2). Model visualization further demonstrated that MOMA attended to lymphocytes, stroma, mucosa, and cancer regions when predicting MSI status (Figures 6C and 6D).

In the revised Results section (page 16, the second paragraph):

MOMA Predicted Copy Number Alterations (CNAs) and Expression Levels of Key Genes in Cancer Development

We further examined the performance of MOMA in predicting copy number alterations (CNAs), whole-genome doubling, and overexpression of the *BECN1* gene using histopathology images. CNAs of many key genes, including *FHIT* and *PTEN*, have been implicated in carcinogenesis. Here we showed that MOMA predicts CNAs in *FHIT* and many other tumor suppressor genes (Figures 7A-C). Compared with PC-CHiP, a commonly used image-based CNA prediction method, MOMA attained substantially improved prediction performance (Supplemental Table 4). In addition to the previously reported histopathology-CNA associations, MOMA further predicted amplifications in *NOL4L*, *HM13*, *FOXS1*, and deletions in *WWOX*, *CCER1*, among many others (Figures 7D-F). Furthermore, MOMA demonstrated improved prediction performance for whole-genome doubling, compared with PC-CHiP (Supplemental Table 5).

In the revised Discussion section (page 21, the third paragraph):

Compared with previously published methods, our approaches achieved substantially improved prediction performance. For instance, we first reproduced a widely used patch-based convolutional neural network¹³ for MSI prediction using the TCGA dataset, and we showed that MOMA achieved a 4%

improvement on the same dataset (Supplemental Table 3). For CNA and WGD prediction, our approaches outperform models derived by the state-of-the-art PC-CHiP methods⁴⁵ by 7-29% (Figure 7). Wilcoxon signed-rank tests confirmed that the performance difference is statistically significant in many clinically important genetic alterations, including *BCL2L1* amplification^{46 47} and *FHIT* deletion⁴⁸ (Supplemental Table 4). Furthermore, we successfully predicted the copy number alterations of 14 additional genes and connected our attention-based deep learning framework with time-to-event models for survival prediction. These methods have the potential to guide clinical decision-making, suggest clinical trial enrollment, and reduce costs attributed to sequencing by serving as a screening tool. We further validated our models in two independent patient populations, i.e., the NHS-HPFS and the PLCO cohorts, which demonstrated the reliability of our approaches when applied to previously unseen populations^{33,49-51}.

The new Supplemental Table 3. Performance comparison between MOMA, a patch-based standard convolutional neural network, and a previously published method (Kather et al.) in MSI prediction.

	MOMA	Patch-based	Kather et al.
Fold 1	0.92	0.85	-
Fold 2	0.92	0.87	-
Fold 3	0.79	0.78	-
Fold 4	0.94	0.94	-
Fold 5	0.89	0.85	-
Mean AUROC	0.88	0.85	0.84
Wilcoxon Signed Rank Test P-Value		Not significant	

The new Supplemental Table 4. Performance comparison between MOMA and PC-CHiP in copy number variation prediction.

	Gene	P-value
Deletion in Colon Cancer	FAT1	Not significant
	PPP2R2A	3.08E-07

	FHIT	7.21E-62
	PTEN	Not significant
	LINC00290	1.80E-136
	MACROD2	Not significant
	CSMD1	Not significant
Amplification in Colon Cancer	BCL2L1	2.71E-168
	ZNF217	Not significant
Deletion in Rectal Cancer	PPP2R2A	2.34E-87
	MACROD2	2.06E-28
	CSMD1	6.15E-54

The new Supplemental Table 5. Performance of whole-genome doubling prediction of MOMA compared with that of PC-CHiP.

	Colon Cancer		Rectal Cancer	
	MOMA	PC-CHiP	MOMA	PC-CHiP
Area Under the Receiver Operating Characteristic Curve	0.72	0.65	0.63	0.51
Wilcoxon Signed Rank Test P-Value	Not significant		5.12E-19	

5. Clinical value of MOMA is insufficiently evaluated. As the MOMA is designed for multiple tasks on clinical practice such as survival and alteration prediction, no data show the clinical value, i.e. can it substitute the current assays used by clinical

platforms? For example, when MOMA is applied to predict the CRC progression (i.e. OS or PFS), how it compares to the known clinical features such staging,

Our response: We thank the reviewer for this point. In our revised Results section, we have demonstrated that MOMA provides additional prognostic prediction beyond the current staging. For example, among stage I or II CRC patients undergoing the standard treatment, MOMA can further differentiate longer-term survivors and shorter-term survivors. In our newly added results, MOMA identified two distinct survival groups among stage III CRC patients as well. These results have demonstrated that MOMA augments, rather than substitutes, the known clinical predictors (e.g., staging) for patient prognosis. Following the reviewer's suggestion, we have revised our Results and Discussion sections to elucidate this point.

In the revised Results section (page 14, the second paragraph):

MOMA Predicted Patients' Overall Survival and Progression-Free Survival

Early-stage colorectal cancer patients have heterogeneous survival outcomes. Although many clinical and molecular predictors have been proposed, they cannot fully explain the divergent prognoses. To address this challenge, we employed MOMA to predict both overall survival and progression-free survival outcomes of stage I-II colorectal cancer patients. Results showed that MOMA successfully identified patients' overall survival outcomes in the TCGA held-out test set (Figure 2A), with a c-index of 0.67 and log-rank test p-value of 0.01 between the two predicted prognostic groups. We further validated our model in two independent external cohorts: NHS-HPFS (Figure 2B; P = 0.0495) and PLCO (Figure 2C; P = 0.046), demonstrating the generalizability of our approaches.

In the revised Results section (page 15, the second paragraph):

Furthermore, we employed MOMA to predict both overall survival and progression-free survival outcomes of stage III colorectal cancer patients. Results showed that MOMA successfully identified patients' overall survival outcomes in the TCGA held-out test set (Figure 4A), with a c-index of 0.66 and log-rank test p-value of 0.02 between the two predicted prognostic groups. We successfully validated our model in two independent external cohorts: NHS-HPFS (Figure 4B; P = 0.0495) and PLCO (Figure 4C; P = 0.04).

In the revised Discussion section (page 21, the second paragraph):

In addition, our stage-stratified survival outcome prediction successfully identified patients with shorter overall and disease-free survival under the standard treatments. Our analyses revealed previously unreported histopathology patterns associated with patient prognosis, which complements clinical staging. This prognostic information will be useful in guiding clinical decision-making. For example, clinicians may provide closer follow-up to patients with suboptimal clinical prognoses, consider more aggressive treatment options, or enroll them in ongoing clinical trials⁴⁴.

6. The authors sometimes describe MOMA as a platform in the manuscript, however, I cannot find a web service or software/portal for potential MOMA users based on code availability.

Our response: We thank the reviewer for this point. To make MOMA maximally accessible to pathologists, oncologists, and biomedical informatics researchers, we have built a new web portal that analyzes any pathology images uploaded by the users and employs our trained model to provide the predictions. This new web portal along with our released software source codes will allow researchers and practitioners to employ our system for pathology evaluation.

In the revised Methods section (page 10, the first paragraph):

To make MOMA easily accessible to pathologists, oncologists, and biomedical informatics researchers, we further developed a web portal (https://rebrand.ly/MOMA_demo) that allows users to upload pathology images and employ our trained models to generate predictions. Our source codes for data analyses and trained models could be found at <https://github.com/hms-dbmi/MOMA>.

7. The 2nd paragraph of introduction lacks the description about related research advances in the CRCs.

Our response: We thank the reviewer for pointing this out. Following the reviewer's suggestion, we have included recent literature related to CRC pathology and added a summary of the key advances and limitations of these research papers in the revised second and third paragraphs of the Introduction section.

In the revised Introduction section (page 4, the second and third paragraphs):

With the recent development of reliable whole-slide pathology scanners and high-performing computer vision techniques, quantitative pathology evaluation has become increasingly feasible⁴. Several studies using machine learning techniques reported remarkable diagnostic accuracy for various cancer types, such as lung, breast, ovarian, renal cell, and colorectal carcinomas⁵⁻¹². Previous works also demonstrated unexpected correlations between histopathology image features and clinically actionable molecular variations, such as microsatellite instability and *PTEN* gene deletion, in colorectal carcinoma samples^{13,14}. These studies indicate that high-resolution pathology images contain underutilized biomedical signals useful for personalizing cancer care¹⁵⁻²⁰.

Nonetheless, many computational challenges hinder the extraction of useful histopathology signals, and several reports expressed concerns about the generalizability of deep learning models²¹. Typical high-resolution digital pathology whole-slide images of colorectal cancer tissues contain up to billions of pixels, making it infeasible for standard convolutional neural networks to process the whole image at once. In addition, deep learning models are highly complex, and it is difficult to connect the image patterns discovered by these data-driven models with biological knowledge²². Furthermore, since there are a large number of parameters that researchers need to optimize in data-driven machine learning models, generalizability to other image acquisition methods remains a substantial challenge to many digital pathology models²³. The lack of extensive validation in different patient cohorts diminished the applicability of machine learning models in clinical settings.

8. Figure 1C/D and related descriptions are not clear, what can readers get from the quantification of microenvironment component?

Our response: We thank the reviewer for this question. To address this issue, we have revised Figures 1C, 1D, and the associated figure legends to clarify the design of our interpretable machine learning approaches and our findings related to each prediction task. Specifically, the revised Figure 1D shows a summary of the regions receiving high attention for overall survival, disease-free survival, and molecular prediction tasks. These results have indicated the relative importance of morphologies in different microenvironments in predicting patient prognoses and molecular profiles.

The revised Figure 1C and the associated figure legend:

C

(C) Model visualization and interpretation. To enhance the interpretability of our machine learning approaches, we compute the importance of each image region to the prediction target by quantifying the performance decay due to occlusion of the region, and we develop a multi-task classification model to quantify the concept (e.g., lymphocyte, stroma, tumor, adipose tissue, mucin, etc.) score using patches whose importance weight is greater than 0.7. This method connects prior histopathology knowledge with quantitative importance metrics independently learned by the models.

The revised Figure 1D and the associated figure legend:

(D) A summary of the pathology concepts associated with survival and multi-omics predictions. The concept scores are plotted on the log scale. OS: overall survival prediction in early-stage CRC; DFS: disease-free survival prediction in early-stage CRC; MSI: microsatellite instability prediction; BRAF: *BRAF* mutation status prediction; BECN: BECN-1 overexpression prediction; CIMP: CpG island methylator phenotype prediction. The major concepts visualized here include lymphocytes (LYM), cancer-associated stroma (STR), tissue debris (DEB), mucus (MUC), smooth muscle (MUS), colorectal adenocarcinoma epithelium

(TUM), and adipose tissue (ADI). The score for each concept indicates the relative importance of each type of microenvironment in predicting patient prognoses or the selected multi-omics variations with clinical implications.

9. It is weird that the author specifically highlight the "BECN1" in the CNV prediction section, please clarify.

Our response: We thank the reviewer for pointing this out. Following the reviewer's suggestion, we have revised the title and narratives of that section to highlight results from both copy number alterations (CNAs) and *BECN1* overexpression to avoid confusion. We have condensed our results in this section to fit the word limit for the journal.

In the revised Results section (page 17, the first paragraph):

MOMA Predicted Copy Number Alterations (CNAs) and Expression Levels of Key Genes in Cancer Development

We further examined the performance of MOMA in predicting copy number alterations (CNAs), whole-genome doubling, and overexpression of *BECN1* using histopathology images. CNAs of many key genes, including *FHIT* and *PTEN*, have been implicated in carcinogenesis. Here we showed that MOMA predicts CNAs in *FHIT* and many other tumor suppressor genes (Figures 7A-C). Compared with PC-CHiP, a commonly used image-based CNA prediction method, MOMA attained substantially improved prediction performance (Supplemental Table 4). In addition to the previously reported histopathology-CNA associations, MOMA further predicted amplifications in *NOL4L*, *HM13*, *FOXS1*, and deletions in *WWOX*, *CCER1*, among many others (Figures 7D-F). Furthermore, MOMA demonstrated improved prediction performance for whole-genome doubling, compared with PC-CHiP (Supplemental Table 5).

Moreover, MOMA revealed the previously unknown correlation between histopathology image patterns and the expression levels of *BECN1* (Supplemental Figure 2A), with the results validated in the NHS-HPFS dataset (Supplemental Figure 2B and Supplemental Table 1). Stratified analyses by colon and rectal cancers showed similar prediction performance in both cancer groups (Supplemental Table 2). The highlighted regions of *BECN1*-high patients have high lymphocyte, mucus, and tumor scores, while the model focused on the stroma, mucus, and tumor regions when evaluating *BECN1*-low patients (Supplemental Figure 2C and Supplemental Figure 2D).

Reviewer #2 (Remarks to the Author): expertise in machine learning and cancer prediction

In this paper, the authors developed a multi-omics multi-cohort assessment (MOMA) platform based on digital pathology images to predict molecular biomarkers and patient survival in colorectal cancer. Strengths of the paper include:

1. Three large patient cohorts (n=1,888)
2. Methodology is state-of-the-art and clearly described.
3. Model visualization and interpretation are a plus

Our response: We thank the reviewer for recognizing the strengths of our study, including the scale, novelty, and interpretability of our Multi-Omics Multi-cohort Assessment (MOMA) platform. Below we answer the reviewer's questions in detail.

Weaknesses include:

1. It is uncertain if proposed framework has any use clinically. First, prediction of patient prognosis may not make any difference in patient management. Second, the performance on using digital pathology images to predict survival and molecular biomarkers is only moderate. For example, c-index for predicting survival is only 0.67. The AUROC for predicting BRAF is only 0.68 in the TCGA test set and 0.66 in the NHS-HPFS cohort. The AUROC for predicting CIMP is only 0.65 for TCGA and 0.64 for the NHS-HPFS. The proposed use for the model is to "provide timely decision support for treatment selection in resource-limiting regions using only the H&E-stained histopathology slides or in settings of limited tissue availability". However, with the current performance, it is uncertain if this is possible. Combination with other modalities such as imaging or clinical variables may be able to achieve better results.

Our response: We thank the reviewer for the questions. Following the reviewer's suggestions, we have revised our manuscript substantially and conducted additional analyses to demonstrate the potential benefits of our prediction models. Below are our answers to the two questions raised by the reviewer.

(1) Our overall survival and progression-free survival prediction models provide additional prognostic signals for patients in the same stage group. Currently, colorectal cancer patients with the same clinical stage and received standard treatments have divergent survival outcomes. We showed that our machine learning-based models can identify additional prognostic signals from the high-

resolution histopathology slides and distinguish longer-term survivors from shorter-term survivors in the same stage groups. Our model can provide useful prognostic information, which can influence treatment decision-making. For example, clinical applications of our prognostic predictions include personalizing the clinical follow-up schedule (e.g., closer clinical follow-up for patients with predicted poorer prognosis), treatment selection (e.g., enrolling patients in clinical trials or considering more aggressive forms of treatments), or advance care planning for patients with poorer prognosis. In addition, we have added a new analysis showing that our methods can be generalized to predict the overall survival and progression-free survival outcomes of stage III colorectal cancer patients. Our results are successfully validated in two independent cohorts (i.e., the Nurses' Health Study-Health Professional Follow-Up Study cohort and the PLCO cohort). For example, we have shown that our overall survival prediction model differentiated longer-term survivors from shorter-term survivors among stage III colorectal cancer patients, with log-rank test P-values of 0.02 in both independent validation cohorts. We have revised our manuscript to reflect these changes and included the new analyses on prognosis prediction for stage III patients.

In the revised Results section (pages 14-15):

MOMA Predicted Patients' Overall Survival and Progression-Free Survival

Early-stage colorectal cancer patients have heterogeneous survival outcomes. Although many clinical and molecular predictors have been proposed, they cannot fully explain the divergent prognoses. To address this challenge, we employed MOMA to predict both overall survival and progression-free survival outcomes of stage I-II colorectal cancer patients. Results showed that MOMA successfully identified patients' overall survival outcomes in the TCGA held-out test set (Figure 2A), with a c-index of 0.67 and log-rank test p-value of 0.01 between the two predicted prognostic groups. We further validated our model in two independent external cohorts: NHS-HPFS (Figure 2B; P = 0.0495) and PLCO (Figure 2C; P = 0.046), demonstrating the generalizability of our approaches. We visualized our models and showed that dense clusters of adenocarcinoma cells are highly indicative of worse overall survival outcomes (Figures 2D and 2E). Analyses that stratified colon cancer and rectal cancer samples show similar prediction performance in both cancer groups (Supplemental Table 2). Quantitative concept-based analyses revealed that regions of carcinoma cells, tumor-associated stroma, and interactions of carcinoma cells with smooth muscle cells in the cancerous regions were related to unfavorable overall survival (Figure 1D).

In addition, MOMA reliably predicted the progression-free survival outcomes of the same cohorts of patients. In the TCGA held-out test set, our progression-free survival outcome prediction model achieved a c-index of 0.88 and a log-rank test p-value of 0.02 in distinguishing the prognostic groups (Figure 3A). We further demonstrated the applicability of our model in the NHS-HPFS cohorts (Figure 3B; c-index = 0.6, $P < 0.005$). When stratifying the datasets into colon cancer and rectal cancer groups, our approaches successfully identified the prognostic differences in both groups (Supplemental Table 2). A sensitivity analysis that was restricted to a surgery-only subgroup demonstrated the robustness of our results (Supplemental Figure 1). Attention visualization showed that morphology patterns in tumor-associated stroma and groups of adenocarcinoma cells are highly indicative of progression-free survival (Figures 3C and 3D). Compared with the overall survival prediction, our progression-free survival model put more emphasis on infiltrating lymphocytes and regions associated with extracellular mucin in its prediction.

Furthermore, we employed MOMA to predict both overall survival and progression-free survival outcomes of stage III colorectal cancer patients. Results showed that MOMA successfully identified patients' overall survival outcomes in the TCGA held-out test set (Figure 4A), with a c-index of 0.66 and log-rank test p-value of 0.02 between the two predicted prognostic groups. We successfully validated our model in two independent external cohorts: NHS-HPFS (Figure 4B; $P = 0.0495$) and PLCO (Figure 4C; $P = 0.04$). On model visualization, we showed that dense clusters of adenocarcinoma cells are highly indicative of worse overall survival outcomes (Figures 4D and 4E). Similarly, MOMA successfully predicted patients' progression-free survival outcomes (Figure 5A), with a c-index of 0.74 and log-rank test p-value of 0.02 between the two predicted prognostic groups in the TCGA held-out test set. These results are validated in our independent external cohorts from NHS-HPFS (Figure 5B; $P = 0.003$). Model visualization showed that dense clusters of adenocarcinoma cells are highly indicative of worse progression-free outcomes (Figures 5D and 5E). Quantitative concept-based analyses revealed that regions of tumor-associated stroma and interactions of carcinoma cells with smooth muscle cells in the cancerous regions were related to unfavorable progression-free survival.

In the revised Discussion section (page 21, the second paragraph):

Our models demonstrated that high-resolution histopathology slides contain robust predictive signals for genetic aberrations and survival outcomes. Since genetic profiling requires abundant tissue samples, additional processing time, and costs, our prediction models can provide timely decision support for

treatment selection in resource-limiting settings using only the H&E-stained histopathology slides or in settings of limited tissue availability. In addition, our stage-stratified survival outcome prediction successfully identified patients with shorter overall and disease-free survival under the standard treatments. Our analyses revealed previously unreported histopathology patterns associated with patient prognosis, which complements clinical staging. This prognostic information will be clinically useful because clinicians can provide closer follow-up to patients with suboptimal clinical prognoses, consider more aggressive treatment options, or enroll them in ongoing clinical trials⁴⁴.

(2) Our results show that MOMA successfully differentiates patients with different overall survival and progression-free survival outcomes. For example, among early-stage colorectal cancer patients, MOMA achieved a c-index of 0.88 and log-rank test P-value of 0.02 in differentiating patients with different progression-free survival outcomes. This result is further validated in our independent validation cohorts (e.g., log-rank test P-value < 0.005 in the NHS-HPFS cohorts). Because patients' survival outcomes are affected by multiple factors, many widely used predictors, including tumor grade, have c-indices less than 0.6. Results from our data-driven analyses exceed these existing predictors and provide further information into the tumor microenvironments (the revised Figure 1D) related to prognosis.

We agree with the reviewer that the AUROC of predicting *BRAF* and CIMP is moderate. Nonetheless, our systematic analyses revealed multi-omics biomarkers with high correlation with histopathology imaging features (e.g., AUROC = 0.88 for predicting MSI status and AUROC = 0.84 for predicting FAT1 deletion) as well as those with weaker associations (e.g., *BRAF* mutation and CIMP). In addition, our approaches outperformed state-of-the-art methods for pathology-based genetic alteration prediction (the new Supplemental Table 4). We agree with the reviewer that combining other imaging (e.g., radiology imaging) and clinical variables may further improve the prediction performance and include this point in our revised Discussion section.

In the revised Results section (page 17, the first paragraph):

MOMA Predicted Copy Number Alterations (CNAs) and Expression Levels of Key Genes in Cancer Development

We further examined the performance of MOMA in predicting copy number alterations (CNAs), whole-genome doubling, and overexpression of *BECN1* using histopathology images. CNAs of many key genes, including *FHIT* and *PTEN*, have been implicated in carcinogenesis. Here we showed that MOMA predicts CNAs in *FHIT* and many other tumor suppressor genes (Figures 7A-C). Compared with PC-CHiP, a commonly used image-based CNA prediction method, MOMA attained substantially improved prediction performance (Supplemental Table 4). In addition to the previously reported histopathology-CNA associations, MOMA further predicted amplifications in *NOL4L*, *HM13*, *FOXS1*, and deletions in *WWOX*, *CCER1*, among many others (Figures 7D-F). Furthermore, MOMA demonstrated improved prediction performance for whole-genome doubling, compared with PC-CHiP (Supplemental Table 5).

In the revised Discussion section (page 21, the third paragraph):

Compared with previously published methods, our approaches achieved substantially improved prediction performance. For instance, we first reproduced a widely used patch-based convolutional neural network¹³ for MSI prediction using the TCGA dataset, and we showed that MOMA achieved a 4% improvement on the same dataset (Supplemental Table 3). For CNA and WGD prediction, our approaches outperform models derived by the state-of-the-art PC-CHiP methods⁴⁵ by 7-29% (Figure 7). Statistical tests confirmed that the performance difference is statistically significant in many clinically important genetic alterations, including *BCL2L1* amplification^{46 47} and *FHIT* deletion⁴⁸ (Supplemental Table 4). Furthermore, we successfully predicted the copy number alterations of 14 additional genes and connected our attention-based deep learning framework with time-to-event models for survival prediction. These methods have the potential to guide clinical decision-making, suggest clinical trial enrollment, and reduce costs attributed to sequencing by serving as a screening tool. We further validated our models in two independent patient populations, i.e., the NHS-HPFS and the PLCO cohorts, which demonstrated the reliability of our approaches when applied to previously unseen populations^{33,49-51}.

In the revised Discussion section (page 23, the second paragraph):

Furthermore, incorporating patients' radiology imaging data, pathology profiles, molecular aberrations, and clinical characteristics may further improve

the prognostic prediction for colorectal cancer patients. Additional research is required to identify the optimal prognostic prediction methods and enable personalized treatments and advance care planning.

2. Reporting the AUROCs alone is not enough for machine learning study to predict binary outcomes. Other performance metrics include accuracy, sensitivity and specificity. For imbalanced outcomes as studied in this paper, precision-recall is a very helpful performance metric to add.

Our response: We thank the reviewer for this suggestion. Following the reviewer’s suggestion, we have added a new table (Supplemental Table 1) in our revised Results section to show these additional performance metrics. Specifically, we provided accuracy, sensitivity, specificity, and precision (using the standard 0.5 cutoff point for the predicted probability) for each task we performed.

The new Supplemental Table 1. Additional model performance metrics for multi-omics characterization via histopathology image analyses.

	Dataset	Accuracy	Precision	Sensitivity (i.e., Recall)	Specificity	AUROC
Microsatellite Instability	TCGA	0.80	0.75	0.89	0.75	0.88
	NHS-HPFS	0.76	0.67	0.86	0.57	0.76
BRAF Mutation c.1799T>A (p.V600E)	TCGA	0.67	0.63	0.78	0.61	0.71
	NHS-HPFS	Mutation Loci Not Available				
BECN1 Overexpression	TCGA	0.60	0.58	0.73	0.61	0.67
	NHS-HPFS	0.85	0.83	0.67	0.64	0.67
CpG Island Methylator Phenotype	TCGA	0.77	0.65	0.68	0.55	0.66
	NHS-HPFS	0.68	0.63	0.67	0.53	0.63

Consensus Molecular Subtypes	TCGA	0.69	0.86	0.73	0.57	0.66
	NHS-HPFS	Transcriptomic Data Not Available				

3. No detail is given on how the genomic data are extracted from the data source (e.g., from the TCGA).

Our response: We thank the reviewer for this point. To address this question, we have added a new paragraph to describe the methods we employed to extract genomic and transcriptomic data from our data sources. These detailed methods will allow researchers to replicate our results in full.

In the revised Methods section (page 6, the first paragraph):

We acquired the digital whole-slide pathology images, whole-exome sequencing results, and RNA-sequencing data of TCGA patients from the National Cancer Institute Genomic Data Commons Portal (<https://portal.gdc.cancer.gov/>). Mutation status, copy number alterations (including genetic amplifications and deletions), microsatellite instability, and CpG island methylator phenotypes (CIMP) of both colon and rectal adenocarcinomas were extracted from the cBioPortal (<https://www.cbioportal.org/>). Whole genome doublings and consensus molecular subtypes (CMS) of colorectal cancers were obtained from a previous TCGA publication²⁹.

4. Colon and rectal cancer are sufficiently different to warrant subgroup analyses rather than grouping together as one cohort.

Our response: We thank the reviewer for this point. In the revised manuscript, we have conducted new analyses that separated colon cancer and rectal cancer in all prediction tasks. We have excluded tasks where the sample sizes are limited (i.e., fewer than 10 patients have death events in survival prediction analyses). Results have shown that we can predict survival outcomes and most of the clinically important genomic variations in both colon cancer and rectal cancer subsets, with AUROC levels comparable to the pooled results. We have included these new results in the revised Results section and the new Supplemental Table 2.

In the revised Results section (page 14, the second paragraph):

Early-stage colorectal cancer patients have heterogeneous survival outcomes. Although many clinical and molecular predictors have been proposed, they cannot fully explain the divergent prognoses. To address this challenge, we employed MOMA to predict both overall survival and progression-free survival outcomes of stage I-II colorectal cancer patients. Results showed that MOMA successfully identified patients' overall survival outcomes in the TCGA held-out test set (Figure 2A), with a c-index of 0.67 and log-rank test p-value of 0.01 between the two predicted prognostic groups. We further validated our model in two independent external cohorts: NHS-HPFS (Figure 2B; P = 0.0495) and PLCO (Figure 2C; P = 0.046), demonstrating the generalizability of our approaches. We visualized our models and showed that dense clusters of adenocarcinoma cells are highly indicative of worse overall survival outcomes (Figures 2D and 2E). Analyses that stratified colon cancer and rectal cancer samples show similar prediction performance in both cancer groups (Supplemental Table 2).

The new Supplemental Table 2. Performance metrics of histopathology-based multi-omics characterization and survival prediction stratified by colon and rectal cancers.

	Stage	Data	C-index	P-value	AUROC
Overall Survival	I & II	TCGA (COAD)	0.73	0.04	-
		TCGA (READ)	0.70	0.04	-
		PLCO (COAD)	0.78	0.04	-
		PLCO (READ)	0.72	0.03	-
		NHS-HPFS (COAD)	0.72	0.03	-
		NHS-HPFS (READ)	0.71	0.04	-
	III	TCGA (COAD)	0.69	0.02	-

		TCGA (READ)	0.73	0.04	-
		PLCO (COAD)	0.70	0.04	-
		PLCO (READ)	0.72	0.04	-
		NHS-HPFS (COAD)	0.65	<0.001	-
		NHS-HPFS (READ)	0.70	0.01	-
Progression-Free Survival	I & II	TCGA (COAD)	0.66	0.04	-
		TCGA (READ)	0.60	0.02	-
		NHS-HPFS (COAD)	0.66	<0.001	-
		NHS-HPFS (READ)	0.62	<0.001	-
	III	TCGA (COAD)	0.75	0.02	-
		TCGA (READ)	0.74	0.02	-
		NHS-HPFS (COAD)	0.73	<0.001	-
		NHS-HPFS (READ)	0.76	<0.001	-
Microsatellite Instability	I - IV	TCGA (COAD)	-	-	0.93
		TCGA (READ)	-	-	0.73
		NHS-HPFS (COAD)	-	-	0.85
		NHS-HPFS (READ)	-	-	0.70
BRAF Mutation (BRAF c.1799T>A)	I - IV	TCGA (COAD)	-	-	0.69
		TCGA (READ)	-	-	-

		NHS-HPFS (COAD)	-	-	-
		NHS-HPFS (READ)	-	-	-
BECN1 Overexpression	I & IV	TCGA (COAD)	-	-	0.67
		TCGA (READ)	-	-	0.71
		NHS-HPFS (COAD)	-	-	0.70
		NHS-HPFS (READ)	-	-	0.68
CpG Island Methylator Phenotype	I - IV	TCGA (COAD)	-	-	0.67
		TCGA (READ)	-	-	-
		NHS-HPFS (COAD)	-	-	0.65
		NHS-HPFS (READ)	-	-	-
Consensus Molecular Subtypes	I - IV	TCGA (COAD)	-	-	0.74
		TCGA (READ)	-	-	0.77
		NHS-HPFS (COAD)	-	-	-
		NHS-HPFS (READ)	-	-	-

In the revised Results section (page 14, the third paragraph):

In addition, MOMA reliably predicted the progression-free survival outcomes of the same cohorts of patients. In the TCGA held-out test set, our progression-free survival outcome prediction model achieved a c-index of 0.88 and a log-rank test p-value of 0.02 in distinguishing the prognostic groups (Figure 3A). We further demonstrated the applicability of our model in the NHS-HPFS cohorts (Figure 3B; c-index = 0.6, $P < 0.005$). When stratifying the datasets into

colon cancer and rectal cancer groups, our approaches successfully identified the prognostic differences in both groups (Supplemental Table 2).

In the revised Results section (page 16, the second paragraph):

Results showed that the AUROC of the TCGA held-out test set is 0.88 ± 0.06 (Figure 6A), and in the NHS-HPFS dataset the AUROC is 0.76 ± 0.04 (Figure 6B and Supplemental Table 1). Our methods improved the AUROC by 4% compared with the state-of-the-art methods by Kather et al¹³ (Supplemental Table 3). In both colon cancer and rectal cancer groups, MOMA showed correlations between histopathology images and MSI status (Supplemental Table 2).

In the revised Results section (page 17, the second paragraph):

Moreover, MOMA revealed the previously unknown correlation between histopathology image patterns and the expression levels of *BECN1* (Supplemental Figure 2A), with the results validated in the NHS-HPFS dataset (Supplemental Figure 2B and Supplemental Table 1). Stratified analyses by colon and rectal cancers showed similar prediction performance in both cancer groups (Supplemental Table 2).

In the revised Results section (page 19, the second paragraph):

To identify the histopathology patterns indicative of the CMS subtypes, we employed MOMA to classify the major CMS subtypes with sufficient numbers of samples (CMS2 and CMS4). Results show that MOMA achieved an AUROC of 0.66 ± 0.04 in the held-out test set not participating in the model development process (Supplemental Figure 8A and Supplemental Table 1). When stratifying the analysis by the colon and rectal cancer groups, we saw a slightly improved performance in CMS prediction (AUROC = 0.74-0.77; Supplemental Table 2).

5. In the prediction of outcome, only stage I and stage II colorectal cancer is used. However, the treatment and prognosis for colon and rectal cancer are very different as well as the work-up. For example, the first-line diagnostic imaging for rectal cancer is MRI and it is frequently treated with radiation before surgery.

Our response: We thank the reviewer for this suggestion. To address this issue, we have conducted new analyses that separate colon cancer and rectal cancer patients into separate datasets, and we have run our data-driven analyses on each set. Results showed that MOMA successfully identified longer-term survivors from shorter-term survivors in both colon and rectal cancer groups in this stratified analysis. We have included these new results in the revised Results section.

The revised sections are included in the response to the preceding comment (comment 4).

6. It is unclear throughout the manuscript what state-of-the-art technique is MOMA being compared to and what the results are.

Our response: We thank the reviewer for this question. In the revised manuscript, we have clarified the state-of-the-art approach (i.e., PC-CHiP for genomic prediction and Kather et al. for MSI prediction) we compared with. In addition, we have compared the performance of our MOMA platform with that of these prior arts using the Wilcoxon rank sum test, and we added new Supplemental Tables 4 and 5 to summarize our results. These results show that our approach has achieved statistically significant improvement in predicting gene amplification and deletion in many clinically relevant genes. We have included these results in the revised Results and Discussion sections.

In the revised Results section (page 16, the second paragraph):

To facilitate the treatment effectiveness prediction for immune checkpoint inhibitors, we employed MOMA to predict the MSI status of each patient. Results showed that the AUROC of the TCGA held-out test set is 0.88 ± 0.06 (Figure 6A), and in the NHS-HPFS dataset the AUROC is 0.76 ± 0.04 (Figure 6B and Supplemental Table 1). Our methods improved the AUROC by 4% compared with the state-of-the-art methods by Kather et al.¹³ (Supplemental Table 3).

In the revised Results section (page 17, the first paragraph):

Compared with PC-CHiP, a commonly used image-based CNA prediction method, MOMA attained substantially improved prediction performance (Supplemental Table 4). In addition to the previously reported histopathology-CNA associations, MOMA further predicted amplifications in *NOL4L*, *HM13*,

FOXS1, and deletions in *WWOX*, *CCER1*, among many others (Figures 7D-F). Furthermore, MOMA demonstrated improved prediction performance for whole-genome doubling, compared with PC-CHIP (Supplemental Table 5).

The revised Figure 7 and legend:

Figure 7. MOMA provides improved copy number alteration prediction compared with the current state-of-the-art methods and predicts additional copy number alterations not achieved in previous studies. We systematically predicted common copy number alterations of colorectal cancer tissues and compared the prediction performance with that of PC-CHIP⁴⁵. The mean and range of AUROC are shown. (A) Prediction of common genetic deletions in patients with colon adenocarcinoma. (B) Prediction of common genetic amplification in patients with colon adenocarcinoma. (C) Prediction of common genetic deletions in patients with rectal adenocarcinoma. (D) Prediction of additional genetic deletions in colon adenocarcinoma. (E) Prediction of additional genetic amplifications in colon adenocarcinoma. (F) Prediction of additional genetic deletions in rectal adenocarcinoma.

The revised Supplemental Table 4:

Supplemental Table 4. Performance comparison between MOMA and PC-CHiP in copy number variation prediction.

	Gene	P-value
Deletion in Colon Cancer	FAT1	Not significant
	PPP2R2A	3.08E-07
	FHIT	7.21E-62
	PTEN	Not significant
	LINC00290	1.80E-136
	MACROD2	Not significant
	CSMD1	Not significant
Amplification in Colon Cancer	BCL2L1	2.71E-168
	ZNF217	Not significant
Deletion in Rectal Cancer	PPP2R2A	2.34E-87
	MACROD2	2.06E-28
	CSMD1	6.15E-54

The new Supplemental Table 5:

Supplemental Table 5. Performance of whole-genome doubling prediction of MOMA compared with that of PC-CHiP.

	Colon Cancer		Rectal Cancer	
	MOMA	PC-CHiP	MOMA	PC-CHiP
Area Under the Receiver Operating Characteristic Curve	0.72	0.65	0.63	0.51
Wilcoxon Signed Rank Test P-Value	Not significant		5.12E-19	

Reviewer #3 (Remarks to the Author): expert in colorectal cancer pathology

This is a well-designed study evaluating the MOMA platform in colorectal cancer patients. The MOMA platform combines histological and genomic data (copy number alterations, gene expression profile and microsatellite instability status). The authors have validated the platform in a large cohort and some of the results are expected or well known (eg.lymphocyte rich population at the invasive margin of the tumour associated with MSI)

The results are quite robust, and this is a very welcome paper in the development of AI in histopathology. However there are several concerns/ limitations in this paper that need to be addressed

Our response: We thank the reviewer for appreciating the rigorous validation, robustness, and novelty of our Multi-omics Multi-cohort Assessment system for quantitative colorectal cancer pathology analyses. Below we answer the reviewer's questions in detail.

1. The study includes stage I and stage II CRC but, particularly stage II, is a very heterogenous group in which risk factors (iv. vascular invasion -mainly extramural-, tumour budding...) play an important role in the prognosis. There is no mention in the paper that these parameters were evaluated. Depending on the presence of some of this risk factors stage II CRC patients might receive chemoradiotherapy and the prognosis (DFS, OS...) will be different in this subgroup. Is there any data on treatment (were all the patients chemo-naive or some patients received adjuvant chemotherapy)

Our response: We thank the reviewer for this question. We agree with the reviewer that differences in treatments due to the differences in the known risk factors can play a role in patients' prognoses. To address this issue, we have conducted a new analysis on stage I and II patients who only received surgery, but not chemotherapy or radiotherapy. Results have shown that our survival prediction methods can predict the progression-free survival outcomes of stage I and II patients who received surgery only. In addition, we have added an additional analysis on the prognostic prediction of stage III patients who have not received neoadjuvant chemotherapy, and we have shown that MOMA can predict progression-free survival in this subgroup of patients as well. Because human-annotated qualitative pathology findings (e.g., vascular invasion) are not reliably reported in all cases in our study cohorts, we focused on the prediction of patient subgroups with relatively homogeneous treatments, as the reviewer suggested. We have added these results to the new Supplemental Figure 1 and the results section of the revised manuscript.

In the revised Results section (page 15, the first paragraph):

When stratifying the datasets into colon cancer and rectal cancer groups, our approaches successfully identified the prognostic differences in both groups (Supplemental Table 2). A sensitivity analysis that was restricted to a surgery-only subgroup demonstrated the robustness of our results (Supplemental Figure 1).

The new Supplemental Figure 1. MOMA predicted survival outcomes of stage I and II colorectal cancer patients receiving surgery only, and stage III cancer patients without neoadjuvant therapy using digital histopathology images. (A) MOMA successfully distinguished the overall shorter-term survivors from longer-term survivors using histopathology images (log-rank test P-value = 0.015) of stage I and stage II patients receiving surgery only. (B) Among stage I and stage II patients without radiotherapy or chemotherapy, MOMA successfully distinguished progression-free survival groups using histopathology images (log-rank test P-value = 0.047). (C) MOMA successfully distinguished the overall shorter-term survivors from longer-term survivors using histopathology images (log-rank test P-value = 0.024) of stage III patients without neoadjuvant therapy. (D) Among stage III patients without neoadjuvant therapy, MOMA successfully distinguished the progression-free survival groups using histopathology images (log-rank test P-value = 0.018).

2. Contrary to the first set of patients /samples studied, the validation cohort included TMAs. This might have affected the results. The authors should explained how the assessed the 10 types of regions in the TMAs. Did TMAs include different areas from the tumour (eg. Core, stroma, invasive front)? or only 1 core per tumour was analysed or only one region per tumour?

Our response: We thank the reviewer for this question. To generate our TMA dataset, we selected one representative core per tumor due to limits in the amounts of available samples and budget. Two experienced colorectal cancer pathologists (J.N. and S.O.) have reviewed the cancer samples and selected the cores to ensure the representativeness of the cores. Thus, our TMA image includes regions of tumor cells, stroma, tumor/stroma interfaces (i.e., microscopic invasive edges), lymphocyte infiltration, and other pathological changes characteristic of the tumor sample from which the core was generated. Our validation using TMA samples has demonstrated the generalizability of our approaches to different imaging modalities for cancer pathology samples.

Following the reviewer's suggestion, we have clarified this point in the revised manuscript.

In the revised Methods section (page 6, the second paragraph):

In addition, we obtained PLCO data from the National Cancer Institute Cancer Data Access System, and we collected clinical, genomic profiles, immunohistochemistry, and hematoxylin-and-eosin (H&E) stained tissue microarray images from the NHS and the HPFS coordinated by Harvard T.H. Chan School of Public Health, Harvard Medical School, and Brigham and Women's Hospital. Notably, colorectal tumor tissue blocks in the NHS and the HPFS were retrieved from over a hundred hospitals throughout the U.S. with variable tissue age, which increased the generalizability of our findings³⁰. For each histopathology from the NHS and HPFS cohort, two experienced colorectal cancer pathologists reviewed the cancer samples and selected the cores to ensure the representativeness of the cores. Thus, the TMA images include regions of tumor cells, stroma, tumor/stromal interfaces (i.e., microscopic tumor invasive edges), lymphocyte infiltration, and other pathological changes characteristic of the tumor sample from which the core was generated. Our multi-center study was approved by the Institutional Review Boards of Harvard Medical School (IRB20-1509).

3. The conclusion on “dense clusters of adenocarcinoma cells being highly indicative of worse OS” (page 13) contradicts current knowledge that tumour with low-stroma (ie high proportion of epithelial cells and low amount of stroma) behave better than tumour with high-stroma content (GW van Pelt 2018; J Gao 2021...). This needs clarification

Our response: We thank the reviewer for this question. These findings do not necessarily contradict the results from van Pelt et al. and Gao et al., because our analyses discovered that our machine learning models leverage pathology patterns in the tumor cell clusters, rather than the relative amount of tumor or stroma in the sample, to predict the overall survival and disease-free survival outcomes. The importance scores shown in Figure 1D indicate the amount of attention our machine learning model paid to different regions of the pathology slides in making the prediction. The scores are not directly related to the amount of area occupied by the tumor or stroma in the slides. These results show that in addition to the known pathology indicators of colorectal patient prognosis, our data-driven approach systematically identified other signals strongly correlated with survival outcomes, and we successfully validated our results in two

independent validation sets (Nurse's Health Study/Health Professionals Follow-up Study and PLCO cohorts). We have revised our Methods sections to elucidate this important point. Following the reviewer's suggestion, we have also clarified our findings in our Results section.

In the revised Methods section (page 12, the second paragraph):

Model Visualization and Interpretation

We developed a novel model interpretation method that incorporates model-derived concept scores and expert-annotated concepts based on prior pathology knowledge. Specifically, we first quantified the importance of each image region by occluding all pixels in the region and computing the extent to which the predicted outcome changed when the region was occluded. We define the importance index of each image region as the numerical change of the predicted probability due to the occlusion of the region. To connect crucial regions with pathology interpretation, we leverage 100,000 histopathology images annotated by gastrointestinal pathologists with seven concepts: colorectal adenocarcinoma epithelium, cancer-associated stroma, lymphocytes, smooth muscle, mucus, adipose tissue, and tissue debris. We developed a deep learning model that classified image regions into these pathology concepts with an accuracy of 99.38%, and we employed this model to compute the concept scores for regions with importance indices greater than 0.7. We scaled our concept score to a range of [0, 100], where 100 indicates the region has the highest relevance to the concept of interest. Thus, our concept scores indicate the amount of attention our machine learning model pays to different regions of pathology changes in making the prediction, and it is not directly related to the amount of area occupied by each pathology pattern in the slides.

In the revised Results section (page 14, the second paragraph):

Quantitative concept-based analyses revealed that regions of carcinoma cells, tumor-associated stroma, and interactions of carcinoma cells with smooth muscle cells in the cancerous regions received high attention when MOMA predicted patients' overall survival outcomes (Figures 1D).

4. The entire discussion around BRAF and response to double/ triple chemotherapy (BEACON study) needs to be properly addressed. The authors claim that MOMA can predict BRAF mutation but not what type of mutation. The only actionable mutation is V600E but around 20% of CRC patients have non-V600E mutations (of uncertain

clinical significance but clearly not associated with poor prognosis or response to encorafenib-cetuximab-binimetinib). Without knowledge of what type of BRAF mutation the data is of very limited value.

Our response: We thank the reviewer for this suggestion. Following the reviewer's suggestion, we have revised Table 1 to summarize the mutational profiles of *BRAF* in our cohorts. Because only the c.1799T>A (p.V600E) mutation (i.e., the V600E mutation) is currently considered clinically actionable, we have conducted a new analysis that predicts the c.1799T>A (p.V600E) mutation, and the results show that our approaches can predict this specific mutation in *BRAF* with an AUROC of 0.71 ± 0.07 (the new Supplemental Figure 3). These new results demonstrate that our platform can predict actionable mutations with direct implications in cancer therapy. As treatment options evolve due to the advent of new targeted therapy drugs, our methods can be tailored to systematically identify pathological signals indicative of actionable mutations in other loci or other genes. We have added these results to the revised manuscript and revised our Methods and Results sections accordingly.

In the revised Methods section (page 8, the second paragraph):

Using the MOMA platform, we conducted multi-omics subgroup predictions on colorectal cancer patients, with a focus on clinically actionable molecular aberrations. Specifically, we predicted the microsatellite instability (MSI) status (65 MSI-high patients; 389 non-MSI-high patients in TCGA), CpG island methylator phenotype (CIMP; 58 CIMP-high tumor patients; 396 CIMP-low/negative tumor patients), *BRAF* c.1799T>A (p.V600E) mutation (48 patients with *BRAF* c.1799T>A (p.V600E) tumor; 529 patients with *BRAF* wild-type tumor), and the most prevalent Consensus Molecular Subtypes (CMS; 152 CMS2 (canonical) patients; 105 CMS4 (mesenchymal) patients) of colorectal cancer. These tasks were constructed as weakly-supervised classification tasks.

In the revised Results section (page 18, the first paragraph):

To identify the morphological impact of clinically important genomic variations, we leveraged MOMA to systematically predict the mutation status of *BRAF*, *HIF1A*, and *PIK3CA*. Results showed that MOMA identified a moderate histopathology signal for predicting *BRAF* c.1799T>A (p.V600E) mutation in the TCGA test set, with an AUROC of 0.71 ± 0.07 (Supplemental Figure 3A and Supplemental Table 1). To further identify the morphological patterns associated with this actionable genetic aberration, we visualized the attention distribution of

our models in Supplemental Figure 3B and Supplemental Figure 3C. The concept scores of mucus, stroma, and tumor regions for *BRAF* mutation with c.1799T>A (p.V600E) detection are 19.89, 18.94, and 16.87, respectively (Figure 1D). When classifying samples with *BRAF* mutation at any loci (n=529) with those without *BRAF* mutation, we also showed that MOMA can identify the morphological signals associated with *BRAF* mutations in general (Supplemental Figures 4A and 4B).

The revised Table 1:

Table 1. Patient characteristics of our study cohorts.

Patient Characteristics		TCGA	NHS-HPFS	PLCO
Number of Patients		N=628	N=927	N=333
Age (Standard Deviation)		66.3±12.8	62.4±9.6	65.0±4.7
Sex	Male	334 (53.2%)	413 (44.6%)	213 (64.0%)
	Female	294 (46.8%)	512 (55.3%)	120 (36.0%)
Race	Not Available	255 (40.61%)	390 (42.07%)	175 (52.6%)
	Black or African American	65 (10.35%)	8 (0.86%)	8 (2.4%)
	White	295 (46.97%)	526 (56.74%)	120 (36.0%)
	Asian	12 (1.91%)	3 (0.32%)	26 (7.8%)
	Native American or Alaska Native	1 (0.16%)	0 (0%)	0 (0.0%)
	Pacific Islander	0 (0.0%)	0 (0%)	4 (1.2%)
Tumor Location	Proximal Colon	258 (42.5%)	469 (50.4%)	127 (38.1%)
	Distal Colon	185 (30.5%)	280 (30.1%)	88 (26.4%)
	Rectum	164 (27.0%)	181 (19.5%)	118 (35.4%)

Disease Stage	Stage I	108 (17.2%)	198 (21.4%)	49 (14.7%)
	Stage II	229 (36.5%)	281 (30.3%)	64 (19.2%)
	Stage III	181 (28.8%)	248 (26.8%)	50 (15.0%)
	Stage IV	90 (14.3%)	134 (14.5%)	16 (4.8%)
	Unknown	20 (3.2%)	66 (7.1%)	154 (46.2%)
MSI	High	65 (14.3%)	150 (16.7%)	-
	Low/negative	389 (85.7%)	750 (83.3%)	-
BRAF mutation	BRAF mutation in any loci	62 (10.4%)	136 (15.0%)	-
	BRAF c.1799T>A (p.V600E) mutation	48 (8.32%)	-	-
	Wild-Type	529 (89.5%)	770 (85.0%)	-
CIMP	High	58 (12.8%)	155 (18.1%)	-
	Low/negative	396 (87.2%)	703 (81.9%)	-

The new Supplemental Figure 3. MOMA identified the association between *BRAF* c.1799T>A (p.V600E) mutation and histopathology image patterns. (A) MOMA characterized a moderate correlation between *BRAF* c.1799T>A (p.V600E) mutation and histopathology image features. Results from the TCGA held-out test set were shown. (B) Attention visualization of a histopathology image from a *BRAF* wild-type patient. (C) Attention visualization of a histopathology image from a *BRAF* c.1799T>A (p.V600E) mutation patient. Regions of muscle, stroma, cancers, and mucus received high attention in this molecular classification task. TUM: colorectal adenocarcinoma epithelium; STR: cancer-associated stroma; MUC: mucus; MUS: smooth muscle.

A**B**

BRAF Wild-Type

**C**

BRAF mutant (V600E only)

REVIEWERS' COMMENTS

Reviewer #1 (Remarks to the Author):

All my concerns have been carefully and properly addressed. I have no further comment.

Reviewer #2 (Remarks to the Author):

The authors have satisfactorily addressed many concerns of the current reviewer including separation of analysis for colon and rectal cancer, addition of other metrics for imbalanced dataset, how genomic data are extracted from source, comparison of current approach with SOTA, etc.

The authors acknowledged that combination with clinical and imaging data may further improve the results and added this to their discussion.

The remaining concerns include the low C-index for overall survival on the external dataset (for example, 0.67 for the TCGA dataset in stage 1-2 and 0.66 in stage III). Also low AUROC in predicting cpG island methylator phenotype and molecular subtypes (AUROC in the 0.6-0.7 range). Although the authors attempt to point out the supposed potential clinical utility of MOMA, it is very uncertain the developed platform will have any role clinically whether now or in the future, especially given equivocal results on the external validation as stated above.

Reviewer #3 (Remarks to the Author):

The authors have addressed adequate most of the concerns and have made amendments in the manuscript.

RESPONSE TO REVIEWERS' COMMENTS

Reviewer #1 (Remarks to the Author):

All my concerns have been carefully and properly addressed. I have no further comment.

Our response: We thank the reviewer for appreciating our additional analyses to address the comments raised previously.

Reviewer #2 (Remarks to the Author):

The authors have satisfactorily addressed many concerns of the current reviewer including separation of analysis for colon and rectal cancer, addition of other metrics for imbalanced dataset, how genomic data are extracted from source, comparison of current approach with SOTA, etc.

The authors acknowledged that combination with clinical and imaging data may further improve the results and added this to their discussion.

Our response: We thank the reviewer for appreciating our stratified analyses, the addition of other performance metrics, details on our data extraction process, and comparisons between our novel methods and the current state-of-the-art methods. As the reviewer pointed out, we have added discussions on future directions on combining clinical and imaging data in the prediction models. Below we answer the reviewer's additional comment in detail.

The remaining concerns include the low C-index for overall survival on the external dataset (for example, 0.67 for the TCGA dataset in stage 1-2 and 0.66 in stage III). Also low AUROC in predicting CpG island methylator phenotype and molecular subtypes (AUROC in the 0.6-0.7 range). Although the authors attempt to point out the supposed potential clinical utility of MOMA, it is very uncertain the developed platform will have any role clinically whether now or in the future, especially given equivocal results on the external validation as stated above.

Our response: We thank the reviewer for pointing out the prediction performance of some of our models included in this systematic study. We agree with the reviewer that our survival prediction model did not perfectly predict the overall survival outcomes of all individuals. In addition, a few molecular prediction tasks included in our analytical plan did not attain an area under the receiver operating characteristic curve (AUROC) that warrants replacements of the current methods for generating ground truth (e.g., genomic sequencing for molecular subtype determination). Nonetheless, our survival prediction framework showed that novel quantitative pathology profiles can provide additional signals that complement known prognostic indicators (e.g., cancer stage) that are used clinically. In addition, we have attained significantly better performance in many molecular prediction tasks (e.g., microsatellite instability, whole-genome doubling, and copy number alterations in genes) compared with the current state-of-the-art methods. Following the reviewer's suggestion, we clarified the clinical implications of our proof-of-concept validation study and identified future research directions that could further improve the prediction performance of these models.

In the revised discussion section: (page 21, the second paragraph)

Our models demonstrated that high-resolution histopathology slides contain useful predictive signals for genetic aberrations and survival outcomes. Because genetic profiling requires additional tissue samples, processing time, and costs, our prediction models that use only the H&E-stained histopathology slides can provide timely decision support for treatment selection in resource-limiting settings or in clinical scenarios with limited tissue availability. In addition, our stage-stratified survival outcome prediction successfully identified patients with shorter overall and disease-free survival under the standard treatments. These results showed that our machine learning approaches extracted stage-independent morphological signals indicative of patients' clinical outcomes. Because patient prognosis depends on many clinical factors, no prediction models can perfectly identify the survival outcomes of individual patients. Nonetheless, our approach unveiled previously unknown histopathology patterns related to patient prognosis, which could be useful in guiding clinical decision-making. For example, clinicians may provide closer follow-up to patients with suboptimal clinical prognoses, consider more aggressive treatment options, or enroll them in ongoing clinical trials⁴⁴.

Reviewer #3 (Remarks to the Author):

The authors have addressed adequate most of the concerns and have made amendments in the manuscript.

Our response: We thank the reviewer for appreciating our extensive amendments and additional analyses included in the previous revision.